# Wind-Wave Characteristics and extremes along the Emilia-Romagna coast

Umesh Pranavam Ayyappan Pillai[1], Nadia Pinardi[1], Ivan Federico[2], Salvatore Causio[2], Francesco Trotta[1], Silvia Unguendoli[3], and Andrea Valentini[3]

[1]Department of Physics and Astronomy, University of Bologna, Bologna, 40127, Italy
[2]Euro-Mediterranean Center on Climate Change, Lecce, 73100, Italy
[3]Hydro-Meteo-Climate Service of the Agency for Prevention, Environment and Energy of Emilia-Romagna, Arpae-SIMC, Bologna, 40122, Italy

*Correspondence to*: Umesh P. A. (umesh.pranavam@unibo.it)

**Abstract.** This study examines the wind-wave characteristics along the Emilia-Romagna coasts (northern Adriatic Sea, Italy) with a 10-year wave simulation for the period 2010-2019 performed with the high-resolution unstructured-grid WW3 coastal wave model. The wave parameters (significant wave height, mean and peak wave period, and wave direction) were validated with the in-situ measurements at a coastal station Cesenatico. In the coastal belt, the annual mean wave heights varied from 0.2-0.4 m, and the seasonal mean was highest for the winter period (> 0.4 m). The Emilia-Romagna coastal belt was characterized by wave and spectra seasonal signals with two dominant frequencies of the order of 10 s/ 5-6 s for autumn and winter, and 7-9 s/4 s for spring and summer. The wavelet power spectra of significant wave height for 10-years show considerable variability, having monthly and seasonal periods. This validated and calibrated data set enabled us to study the probability distributions of the significant wave height along the coasts and define a hazard index based on a fitted Weibull probability distribution function.

## 1 Introduction

The wind induced stress on the sea surface gives rise to wind-waves that affect human activities on the coasts (Armaroli et al., 2019). The prevailing wind-waves of a region determine the defence performance of coastal and offshore structures, and therefore a precise information on wind-waves is crucial for coastal operations and defence systems. During extreme events, the wind-waves modify the total water-level elevation, leading to a higher risk of overtopping which can damage infrastructures. The Intergovernmental Panel on Climate Change (IPCC, 2007) has also highlighted the need for a long term evaluation of wind-wave climate trends for coastal resilience (Hemer et al., 2012). IPCC (2021) indicates the necessity of a regional evaluation of climate change, with various target factors that can aid in risk management and policy making. The report points out that over the 21[st] century nearshore regions will encounter sea level rise, thereby adding to more persistent coastal flooding (across low lying regions) and associated coastal erosion.

Over the globe, wave climatology studies using reanalysis datasets and model hindcasts are reported by Carter et al. (1991), Sterl et al. (1998), Young (1999), Cox and Swail (2001), Sterl and Caires (2005), Hemer et al. (2010), Semedo et al. (2011), Young et al. (2011), Zheng et al. (2016), and De Leo et al. (2020). Wind speed and wave height climatologies with emphasis on the Southern Ocean is described in the works of Young (1999), Young and Holland (1996), Young and Donelan (2018).

Past studies on regional scales (Young et al., 2020) based on observations and numerical modelling were also reported by various researchers on different regions such as: Northern Hemisphere (Woolf et al., 2002; Reistad et al., 2011); Southern Hemisphere (Gorman et al., 2003); Mediterranean Sea (Lionello and Sanna, 2005; Lionello, 2012; Clementi et al., 2017; Ravdas et al., 2018; Morales-Márquez et al., 2020; De Leo et al., 2021; Barbariol et al., 2021; Amarouche et al., 2022), Persian Gulf (Kamranzad et al., 2013), western Australia (Bosserelle et al., 2012), eastern North Atlantic (Dodet et al., 2010), southeast

Pacific ocean (Aguirre et al. 2017), Indian Ocean (Stopa and Cheung, 2014), Black Sea (Akpinar and Komurcu, 2013; Arkhipkin et al., 2014; Fedor et al., 2020), and China Seas (Zheng and Li, 2015; Qian et al., 2020).

Numerous studies have been reported for the Adriatic Sea, using numerical models to demonstrate the wind-wave climate characteristics. In the Adriatic there are many wind-wave forecast systems, including the Henetus forecast system described in Bertotti et al. (2011). Other state-of-the-art models include the Nettuno system as reported in Bertotti et al., (2013) which

combines the atmospheric model COSMO (Steppeler et al., 2003) and the wave model WAM (Komen et al. 1994) and SWAN-MEDITARE which combines COSMO with SWAN (Valentini et al., 2007; Russo et al., 2013). Donatini et al. (2015) also have implemented high resolution model chains for wind-wave forecasting in the Mediterranean and Adriatic Seas, which uses a combination of the atmospheric model WRF and wave model MIKE-21 (DHI, 2017). In a study over the Gulf of Taranto in southern Italy, a multi-nesting approach was adopted to evaluate coastal wave dynamics and hydrodynamics (Gaeta et al.,

2016). In the Adriatic Sea, Sikiric et al. (2018) implemented the unstructured-grid WAVEWATCH-III (WW3) (WW3DG, 2016) with 2 km wind forcings from ALADIN forecasts (Farda et al., 2007). The study showed a good match with satellite measurements (SARAL) as compared to CryoSat-2 and Jason-2. The results were in agreement with the studies by Sepulveda et al. (2015) which showed that SARAL estimates of wave heights were far better than CryoSat-2 and Jason-2. Cavaleri et al. (2018) also reported on the application of SARAL data, producing good results.

In a study of the northern Adriatic, Lionello et al. (2012) used the WAM model to predict extreme wind-waves and the associated storm surge effects. In the Adriatic a modelling combination of WAM + SHYFEM (Komen et al., 1994; Umgiesser et al., 2014) forced with analysis and forecast ECMWF winds was used to forecast the 2018, October 29 storm (Cavaleri et al., 2019) conditions in northern Italy. The application of corrected forecast winds (ECMWF) within these models provided consistent results in line with measurements. High waves in the northern Adriatic Sea were reported in a recent study by

Cavaleri et al. (2021).

Studies by Katalinic et al. (2015) reported that in the Adriatic basin, the wind speed and wave height increase from the northern to the southern areas with a maximum mean (annual) Hs of 0.68 m. These results are underestimated as compared with the findings of Queffeulou & Bentamy (2007), resulting from a 14-year (1992-2005) satellite mission that revealed a mean Hs of 0.85 m. Queffeulou & Bentamy also showed that in the Adriatic Sea, 80% of the Hs were lower than 1.10 m. An intercomparison of WAM and WW3 models in the Adriatic and North Sea, based on testing various input physics, was reported by Benetazzo et al. (2021). The analysis aided in investigating the processes that lead to the generation of higher waves in the context of storms.

In the light of several hazardous and extreme events in the Emilia-Romagna (ER) coastal area, several studies have investigated: (i) coastal risk and vulnerability to flooding, and erosion (Armaroli et al., 2009; Sekovski et al., 2015; Armaroli and Duo, 2018; Sanuy et al., 2018; Armaroli et al., 2019; Ferrarin et al., 2020), (ii) sea level rise, land subsidence, and littoral hydrodynamics (Perini et al., 2017; Gaeta et al., 2018), (iii) identification of storm thresholds (Armaroli et al., 2012), and (iv) forecasting of coastal flooding (Biolchi et al., 2020; 2021).

To the best of our knowledge, no studies have been carried out to date on the wind and wave characteristics and extremes in the ER coastal belt with high resolution wind-wave models. Our study focusses on the prevailing wind-wave climatology in the coastal belt of the ER (northern Adriatic Sea) for a period of 10 years (2010-19), the characterization of the wind wave regimes and the study of extreme wave conditions along the coastal belt to quantitatively determine the extreme wave hazard. We use a specific probability distribution function fitting procedure to the wind wave model data and thereby extract hazard indices for different coastal points. We believe that our 10-year model simulation with appropriate validation at a coastal location will be useful for hazard estimations along the ER coastal area. For the first time we discuss the probability distribution of waves that are essential to quantify the extremes and their hazard.

The paper is organized as follows. Section 2 outlines the study area. Section 3 describes the wind wave model used in the study, the model forcing, and the validation buoy data used. Section 4 describes the wind and wave climate in the ER coastal belt together with the wave spectra characteristics and wavelet analysis. Section 5 presents the analysis of the probability density distribution and the hazard index for extreme events. Finally, Section 6 summarize the key findings from the study with a brief conclusion.

## 2 The study area - the Emilia-Romagna coast

The study area is the coastal waters of Emilia-Romagna, situated in northern Italy along the Adriatic Sea, with a coastline including natural zones and dunes to long stretches sheltered by groynes and breakwaters (Armaroli et al., 2012). The coastline is 130 km long (Harley et al., 2016) with the Po delta as the northern boundary and the town of Riccione at the southernmost point. Fig. 1 shows the study area in the ER coastal belt.

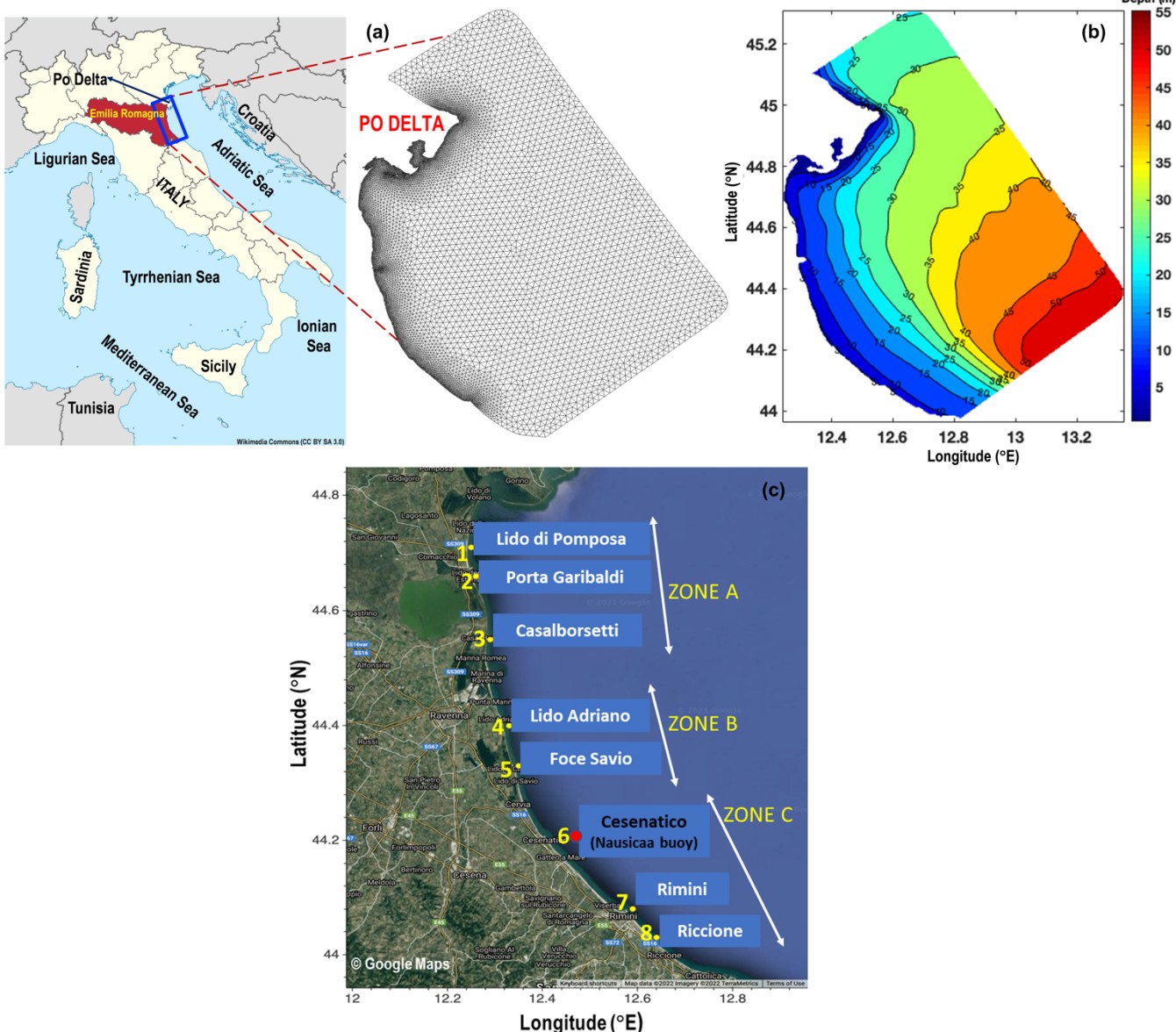

**Figure 1.** (a) The Emilia-Romagna coastal belt with the unstructured mesh, (b) bathymetry for the model domain, and (c) the control points across the coastal belt used for analysis and validation. The Nausicaa buoy in Cesenatico (at the station 6), was used in this study to validate the hindcast wave parameters.

There are two main wind patterns in this region – the Bora and Scirocco winds (Pandzic and Likso, 2005; Umgiesser et al., 2021). Severe wind storms occur from the east-northeast, i.e. the prevailing direction of the Bora winds. The Sirocco winds are associated with low pressure systems over the Italian peninsula and the Ionian Sea. Owing to the restricted fetch, i.e., limited extension of the wave generation area, the Bora winds generate young, and steep waves that breaks frequently (Cavaleri

et al., 1991), while the Sirocco winds generate longer fetch waves across the Adriatic Sea (Cavaleri, 2000). Thus the swell seas are controlled by the Sirocco winds and the seas are dominated by the Bora winds (Bonaldo et al., 2017).

The ER coastal area is subdivided into three major zones (Fig. 1c) which correspond to different coastal trophic conditions (Fiori et al., 2016). The station locations are the land town locations perpendicular to which the environmental agency monitoring transects are done monthly and weekly to monitor the marine ecosystem good environmental status. Thus, knowing the prevailing winds and waves at these locations could be of importance for the management of this important coastal area.

The prevailing hydrodynamics, show that the region is microtidal with spring tides (80-90 cm), and neap tides (30-40 cm), with strong diurnal and semi-diurnal components (Armaroli and Duo, 2018). A low energy wave climate (IDROSER, 1996; Ciavola et al., 2017) has been reported along the coastal belt of ER, i.e., 60% Hs < 1 m. Armaroli et al. (2012) reported that waves originating from east correspond to a proportion of 91% Hs < 1.25 m, owing to the controlled fetch.

## 3. Numerical wave model set up

In this study, the third-generation unstructured-grid spectral-wave model, WW3 (version 5.16, WW3DG, 2016) was used to evaluate the nearshore waves. WW3 is a universally accepted wave model (Tolman et al., 2002) with continuous updates of ocean wave physics. The model is formulated by solving the action-density, balance equation:

$$\frac{\partial N}{\partial t} + \frac{1}{cos\phi}\frac{\partial}{\partial \phi}\dot{\phi}Ncos\theta + \frac{\partial}{\partial \lambda}\dot{\lambda}N + \frac{\partial}{\partial k}\dot{k}N + \frac{\partial}{\partial \theta}\dot{\theta}_g N = \frac{S}{\sigma} \tag{1}$$

The left-hand side of equation (1) denotes the changes in wave action density (i.e., local rate), generation in physical space, shifting of action density (frequency/ direction), owing to spatio-temporal changes in depth, and current. $\lambda$- denotes longitude, $\varphi$- latitude, $\theta$- direction of wave propagation, $k$- wave number, $\sigma$ and $t$ represents the intrinsic angular frequency, and time respectively. The source term, $S$ in (1), used in this paper is both the wind-input and dissipation source package ST4 (Ardhuin et al., 2010) or ST6 (Zieger et al., 2015; Rogers et al., 2012; and Babanin, 2011), the bottom friction JONSWAP parameterization (Joint North Sea Wave Project) (Hasselmann et al., 1973) or SHOWEX (Shoaling Waves Experiment) formulation (Ardhuin et al., 2003) for sandy bottoms. In the section on sensitivity experiments we used a combination of these source terms.

The WW3 model grid (Fig. 1) is divided into 15392 elements, linked with 8148 nodes, with a resolution of about 300 m at the coast, and 2.5 km at the open boundary (Fig. 1a). The merged European Marine Observation and Data Network (EMODnet) data (250 m resolution) and multibeam high-resolution measurements from Arpae (Regional Agency for Prevention, Environment and Energy of Emilia Romagna) serves as the bathymetry of the ER domain (Fig. 1b). The model spectrum is

sampled in 24 directions and 30 frequencies (0.0500-0.7932 Hz), with an increment factor of 1.1. The model time steps are set as: (i) maximum global time step: 200 s, (ii) maximum CFL (Courant-Friedrich-Levy) time step X-Y: 50 s, (iii) maximum CFL time step k-theta: 50 s, and (iv) minimum source term time step: 10 s. The source term for linear input and wind input uses the parameterization formulated by Cavaleri and Malanotte-Rizzoli (1981), and Donelan et al. (2006). The Generalized

Multiple DIA (GMD), was used to simulate the non-linear interactions (Tolman 2010, 2013, 2014), the dissipation physics were based on Rogers et al. (2012), and the SHOWEX formulations by Ardhuin et al. (2003) were used to simulate the bottom friction. The SHOWEX parameterisation is ripple-induced bottom friction, which considers the formation of sand ripples on the bottom. Breaking (depth-induced) is activated using the Battjes and Janssen (1978) physics.

The WW3 model is forced every six hours with the ECMWF analysis winds at 0.125° horizontal resolution. The model winds were validated at three stations, namely Porto Corsini (44.49°N, 12.28°E), Porto Garibaldi (44.67°N, 12.24°E) and Cesenatico Port (44.20°N, 12.40°E) along the ER coastal belt. The wind speed comparison statistics as indicated in Table 1 showed correlations of the order 0.7, with bias of -0.2 m/s indicative of underestimation of wind speed, and RMSE of 1.8 m/s. Larger biases of the order of -0.6 m/s and correlations as low as 0.5 exist during summer and some autumn seasons.


**Table 1.** Quality assessment of ECMWF winds with observed wind speeds for selected stations.

| Statistics | Wind speed (m/s) | | | | |
|---|---|---|---|---|---|
| | (a)  Porto Corsini [Year: 2013] | | | | |
| | Full year | Winter | Spring | Summer | Autumn |
| R | 0.7 | 0.7 | 0.7 | 0.5 | 0.7 |
| Bias | -0.2 | 0.2 | -0.1 | -0.3 | -0.4 |
| RMSE | 1.8 | 1.8 | 1.9 | 1.6 | 2 |
| | (b)  Porto Garibaldi [Year: 2018] | | | | |
| R | 0.7 | 0.8 | 0.7 | 0.5 | 0.8 |
| Bias | -0.2 | 0.2 | -0.3 | -0.5 | 0 |
| RMSE | 1.8 | 1.7 | 1.6 | 1.9 | 1.9 |
| | (c)  Cesenatico Port [Year: 2015] | | | | |
| R | 0.7 | 0.8 | 0.8 | 0.5 | 0.6 |
| Bias | -0.2 | 0 | -0.3 | -0.6 | 0.2 |
| RMSE | 1.9 | 1.7 | 2 | 1.9 | 2 |
| *R: Correlation, **RMSE:** Root Mean Square Error* | | | | | |

The wave lateral boundary values are provided by the Copernicus Marine Environment Monitoring Service-CMEMS model (https://marine.copernicus.eu/, Korres et al., 2021) at a resolution of ~ 4.5 km hourly. The open boundary nodes are forced via JONSWAP wave spectrum approximation (Yamaguchi, 1984) based on the CMEMS wave parameters (significant wave height, peak period, and mean direction).


### 3.1. Wave observational dataset and validation method

In order to validate the model hindcasts, we used the wave buoy Nausicaa in Cesenatico (44.21°N, 12.47°E, Station 6) as shown in Fig. 1c. This station is situated away from the coast of Cesenatico municipality, and is supported with a Datawell Directional Wave Rider (MkIII-70 wave) buoy, called Nausicaa (https://www.arpae.it/it/temi-ambientali/mare/dati-e-indicatori/dati-boa-ondametrica) which has been maintained by Arpae since 23 May 2007. The location of the buoy is 8 km offshore Cesenatico, at a depth of approximately 10m, in a region inaccessible to fishing, navigation, and moorings. Wave

data such as height (Hs), period and direction of waves every 30 minutes constituted the basic validation data set for the modelling period from January 2010 to December 2019.

Wave model parameters such as wave height, period, and direction were extracted and analyzed for eight control points as shown in Fig. 1c. The details of the control points are described in Table 2. The model simulated 1D wave spectra are extracted and analyzed based on seasons.


**Table 2.** Details of the control points 1 to 8.

| Control points | Station Name | LON (°E) | LAT (°N) | Depth (m) | ZONE |
|---|---|---|---|---|---|
| 1. | Lido di Pomposa | 12.25 | 44.71 | 5.8 | A |
| 2. | Porto Garibaldi | 12.26 | 44.66 | 5.1 | A |
| 3. | Casalborsetti | 12.29 | 44.55 | 5.0 | A |
| 4. | Lido Adriano | 12.33 | 44.40 | 7.7 | B |
| 5. | Foce Savio | 12.35 | 44.33 | 5.3 | B |
| 6. | Cesenatico | 12.47 | 44.21 | 10.4 | C |
| 7. | Rimini | 12.59 | 44.08 | 8.1 | C |
| 8. | Riccione | 12.64 | 44.03 | 6.2 | C |

The skill of the model to reproduce the observations at the Nausicaa buoy location was assessed by standard statistics namely: correlation coefficient (R), bias, and root mean square error (RMSE):

$$R = \frac{\sum_{i=1}^{n}(P_i - \bar{P})(O_i - \bar{O})}{\sqrt{\sum_{i=1}^{n}(P_i - \bar{P})^2(O_i - \bar{O})^2}} \qquad (2)$$

$$Bias = \frac{1}{n}\sum_{i=1}^{n}(P_i - O_i) \qquad (3)$$

$$RMSE = \sqrt{\frac{1}{n}\sum_{i=1}^{n}(P_i - O_i)^2} \qquad (4)$$

where, model estimates are denoted by 'P', 'O' represents observational data, 'n' indicates number of data points, and overbar denotes mean values.


## 3.2 Sensitivity experiments for wave model parametrizations

Three sets of sensitivity experiments using WW3 were executed using a combinations of wave physics:

(i) ST4 + JONSWAP (EXP1),

(ii) ST4 + SHOWEX (EXP2), and

(iii) ST6 + SHOWEX (EXP3),

for the representative months of February (winter) and September (autumn) 2018. The combination of ST6 with JONSWAP is not considered because this bottom friction is not suitable for sandy beaches as already the EXP1 will show.


**Table 3.** Skill scores for the sensitivity experiments.

| Experiment | Significant wave height (Hs in m) | | | | | |
| --- | --- | --- | --- | --- | --- | --- |
| | February 2018 | | | September 2018 | | |
| | R | Bias (m) | RMSE (m) | R | Bias (m) | RMSE (m) |
| EXP1 | 0.93 | -0.12 | 0.29 | 0.92 | -0.15 | 0.21 |
| EXP2 | 0.91 | -0.09 | 0.32 | 0.90 | -0.12 | 0.18 |
| EXP3 | 0.94 | -0.04 | 0.26 | 0.96 | -0.09 | 0.14 |
| *R: Correlation, **RMSE**: Root Mean Square Error* | | | | | | |

The Hs comparison with the Nausicaa buoy is shown in Table 3 which highlights that the best physics is given by EXP3. The mean buoy Hs is 0.92 m and 0.38 m for February and September 2018 respectively. The comparison of the mean wave period, Tm (not shown), for the three experiments showed a higher performance using the combination of ST6+SHOWEX. The sensitivity study produced sufficient confidence in using the ST6+SHOWEX physics for the ER region. This wave physics was thus adopted for the 10-year simulation.

## 3.3 Validation of wave hindcasts


The model outputs, such as significant wave height (Hs), mean wave period (Tm), peak wave period (Tp), and mean wave direction ($\theta$m), were compared with the buoy observations for the ten-year period 2010-2019. Fig. 2 shows the 10-year comparison of Hs, which qualitatively demonstrates that the overall model Hs followed the buoy values also in peak events at the Cesenatico station (station 6 in Fig. 1c). This is a consistency check of model against observations as required for "goodness" indicators in numerical weather predictions (Murphy, 1993). The model also captures the seasonal variations at the coastal location. In general, the lower Hs values are slightly overestimated, while higher Hs are underestimated.

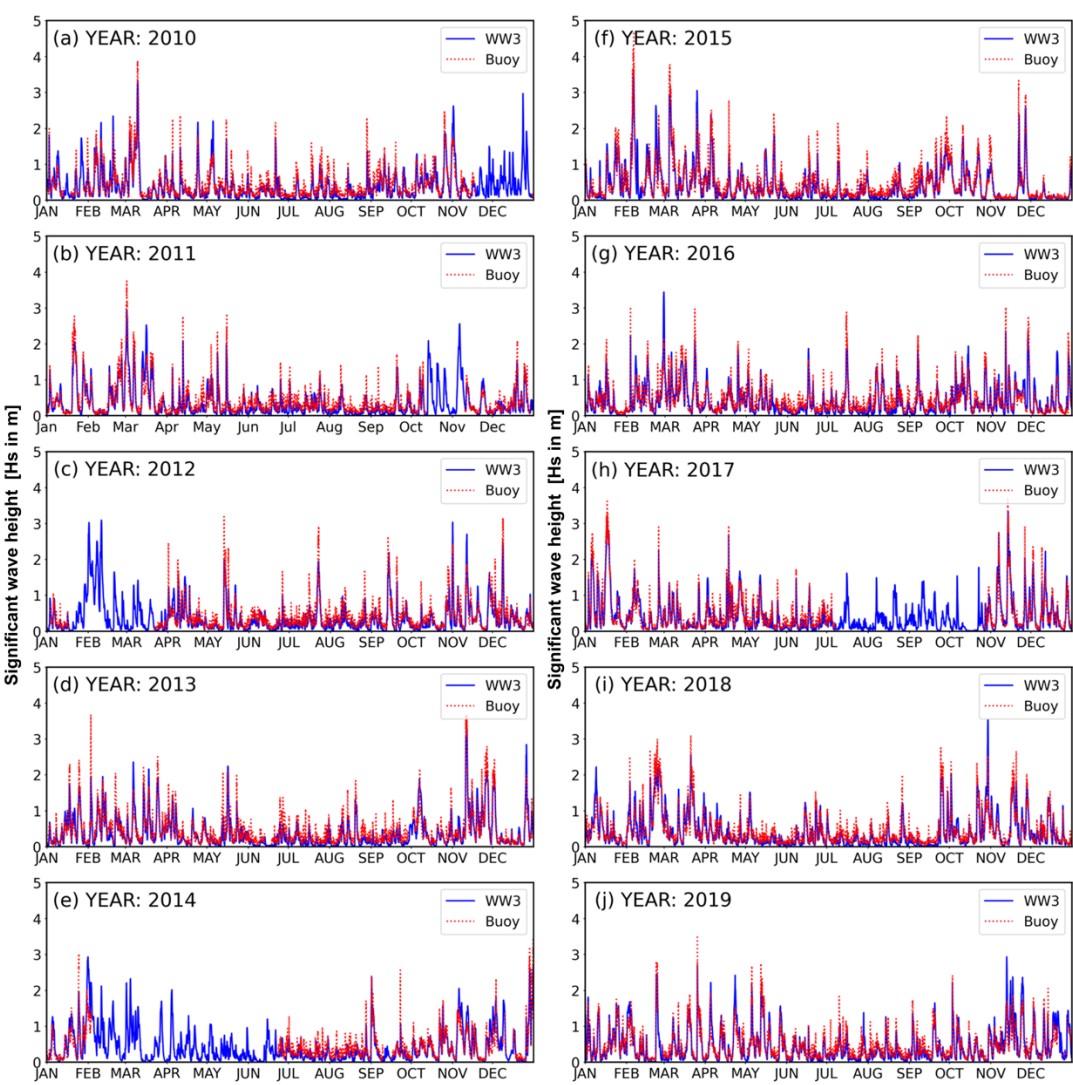

**Figure 2.** Time series plot of (a-j) significant wave height (in metres, indicated by blue solid lines) and observations (red dotted lines) for 2010-19 at station 6 (Cesenatico, see Fig. 1c for location).

**Table 4.** Statistics of the comparison of buoy measurements (Cesenatico, Station 6) with model results for 2010-2019.

| Statistics | 2010 | 2011 | 2012 | 2013 | 2014 | 2015 | 2016 | 2017 | 2018 | 2019 |
|---|---|---|---|---|---|---|---|---|---|---|
| **Significant wave height (Hs in m)** | | | | | | | | | | |
| **R** | 0.882 | 0.903 | 0.876 | 0.814 | 0.860 | 0.917 | 0.890 | 0.932 | 0.915 | 0.897 |
| **Bias** | -0.055 | -0.066 | -0.076 | -0.065 | -0.022 | -0.053 | -0.045 | -0.031 | -0.035 | -0.016 |
| **RMSE** | 0.211 | 0.193 | 0.205 | 0.211 | 0.252 | 0.209 | 0.201 | 0.194 | 0.193 | 0.206 |
| **Mean wave period (Tm in s)** | | | | | | | | | | |
| **R** | 0.718 | 0.776 | 0.724 | 0.739 | 0.809 | 0.746 | 0.740 | 0.709 | 0.746 | 0.777 |
| **Bias** | -0.23 | -0.371 | -0.321 | -0.255 | -0.112 | -0.363 | -0.194 | -0.018 | -0.159 | -0.071 |
| **RMSE** | 0.911 | 0.797 | 0.841 | 0.872 | 0.828 | 0.904 | 0.804 | 0.838 | 0.854 | 0.821 |
| **Peak wave period (Tp in s)** | | | | | | | | | | |
| **R** | 0.653 | 0.530 | 0.598 | 0.621 | 0.705 | 0.543 | 0.603 | 0.605 | 0.653 | 0.642 |
| **Bias** | -0.305 | -0.255 | -0.325 | -0.273 | -0.258 | -0.382 | -0.183 | 0.151 | -0.084 | 0.079 |
| **RMSE** | 1.618 | 1.782 | 1.611 | 1.687 | 1.483 | 1.860 | 1.575 | 1.636 | 1.597 | 1.582 |
| *R: Correlation,* **RMSE:** *Root Mean Square Error* | | | | | | | | | | |

Table 4 shows the validation statistics for each year. The mean of model/ buoy estimates for Hs, Tm, and Tp are 0.40 m/ 0.45 m, 3.02 s/ 3.23 s, and 4.17 s/ 4.36 s respectively. On an average the model underestimates the measurements (as seen from the

 negative bias for most of the years). A high correlation is shown ranging from 0.81 to 0.93 for 2010-19, with the highest correlation for 2017. The Tm comparison revealed a lower correlation of the order 0.72 to 0.81, compared to Hs. The negative bias (-0.371 to -0.018 s) indicated an underestimation of Tm, with a corresponding RMSE of the order 0.79 s to 0.91 s. Similarly, the Tp also showed a lower correlation (0.53 to 0.70) in comparison to Hs and Tm. Tp also showed underestimations as revealed from the bias of the order -0.382 to 0.151 s, with an RMSE varying from 1.48 to 1.78 s.

Figure 3 represents the observations-model scatter plot of Hs for the period 2010-19 (Fig. 3a), and the seasonal scatter as shown in Fig. 3(b-e) for the station 6. The red dashed line denotes the best data fit for the comparison. The comparison of Hs for 2010-19 (Fig. 3a) shows that there is relatively good agreement between model Hs and measurements with a high correlation of 0.90. There is a slight underestimation (Bias= -0.05 m), with an RMSE=0.21 m. The seasonal scatters for winter,

 spring, and autumn (Figs. 3b, c, e) showed high correlations, with a slight underestimation in relation to buoy observations. The summer seasons (Fig. 3d) showed a comparatively lower correlation with an underestimation of Hs. In general, the model Hs, underestimates the buoy data, specifically the higher Hs, and similar underestimations have been reported in many past studies such as Ardhuin et al. (2007), Korres et al. (2011), and Clementi et al. (2017).

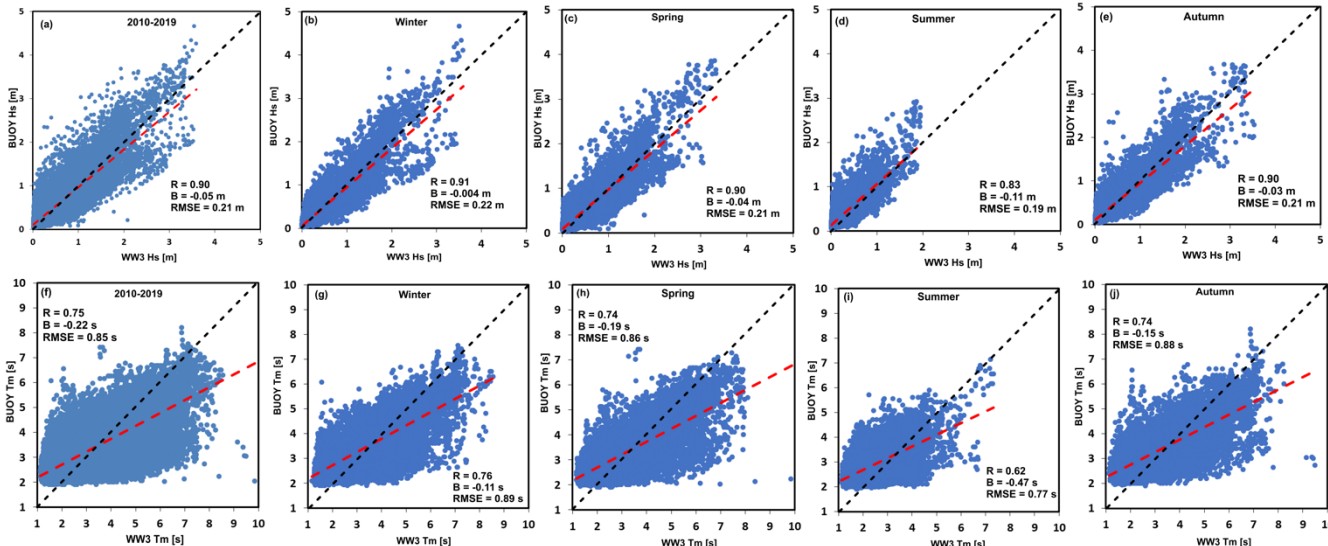

**Figure 3.** Observations-model scatter plot of Hs (in m) for (a) 2010-19, (b) winter, (c) spring, (d) summer, and (e) autumn (top panel) at station 6 (Cesenatico, see Fig. 1c for location). The bottom panel shows scatter plots of mean wave period (Tm in seconds) for (f) 2010-19, (g) winter, (h) spring, (i) summer, and (j) autumn. Black dashed line indicates the best fit (1:1 slope), and dashed red line represents the data fit. *[R: Correlation, B: Bias, and RMSE: Root Mean Square Error].*

The comparison of Tm for 2010-19 is shown in Fig. 3(f), and for the seasons in the Fig. 3(g-j), revealing a larger scatter in comparison to Hs. During 2010-19 (Fig. 3f), the simulated Tm is lower than the buoy measurements and shows a lower performance (R=0.75) in comparison to Hs. The winter, spring, and autumn seasons (Fig. 3g, h, j) showed a moderate correlation of 0.74 to 0.75, while the lowest correlation was observed in summer (0.62). For all the seasons, underestimations of Tm were noted, with the maximum in summer (B=-0.47s), and lowest in winter (B=-0.11s).

## 4. Characterization of the Emilia-Romagna wind and wave fields

### 4.1 Wind climatology of the Emilia-Romagna coast

Below we present the wind climatology in the ER region based on the ECMWF analysis winds over a period of 10 years. The seasons are presented as: winter (Dec-Jan-Feb), spring (Mar-Apr-May), summer (Jun-Jul-Aug), and autumn (Sep-Oct-Nov).

### 4.1.1. Climatology of wind speed and direction

The analysis of wind speed and direction over the ER coast for the period 2010-2019 is presented in Fig. 4. The annual mean characteristics showed a very precise pattern, with the winds reaching the coast from the east-northeast. The annual mean wind speeds were of the order 0.5-2m/s, with a large standard deviations (SD) of 1.6-3 m/s.

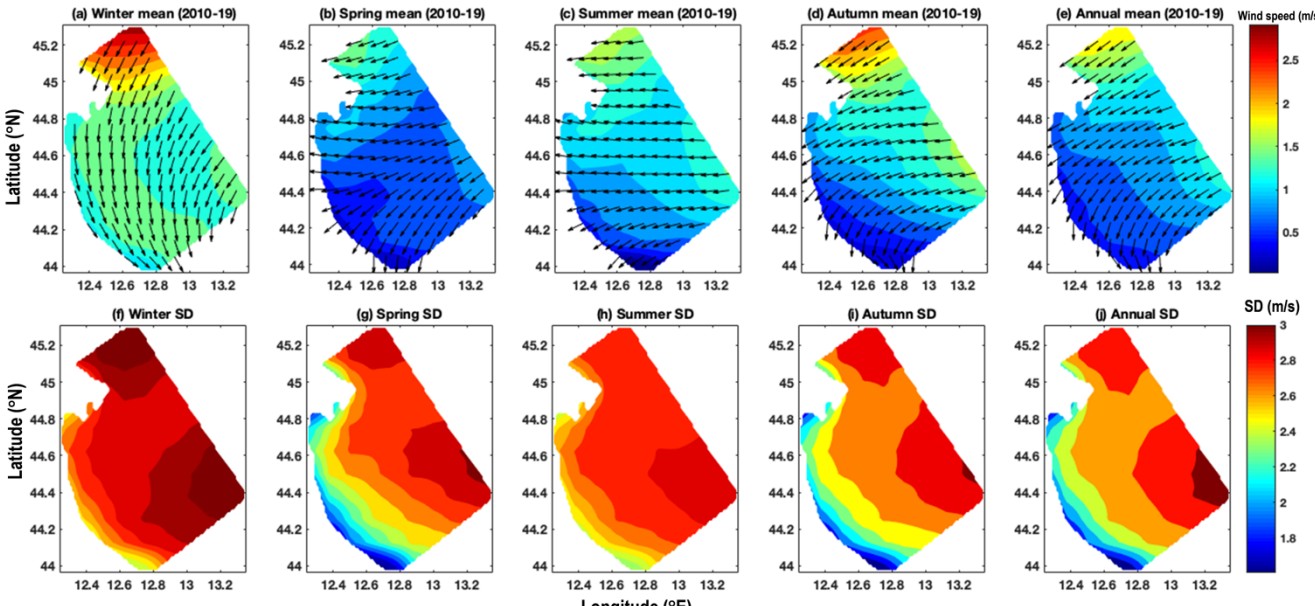

**Figure 4:** Wind climatology for the Emilia-Romagna region based on ECMWF analysis wind data for 2010 to 2019. Mean wind speed and direction for (a) winter, (b) spring, (c) summer, (d) autumn, and (e) annual (top panel). The lower panel shows the standard deviation (SD) of wind speed for (f) winter, (g) spring, (h) summer, (i) autumn, and (j) annual periods.

The lowest wind speeds were observed during spring and summer (1.5/1.8 m/s), followed by autumn (2.4 m/s), and with highest wind speeds (2.9 m/s) during winter. Overall, for the winter and spring the approaching wind is easterly related to the Bora wind climatological direction. In the summer, the mean wind direction is from the southeast, owing to Sirocco events. The spatial distribution of seasonal and annual SD of wind speed from 2010-19 is shown in the bottom panel of Fig. 4(f-j). The annual SD varies from 1.6 to 3 m/s in the entire domain (Fig. 4j), and the annual maximum is further offshore from the ER coastal belt. During the winter (Fig. 4f), the SD varies from 2.2 to 3 m/s, and in spring, from 1.2-3 m/s (Fig. 4g). While in summer and autumn, the SDs are 2.2-2.6 m/s, and 1.6-3 m/s respectively.

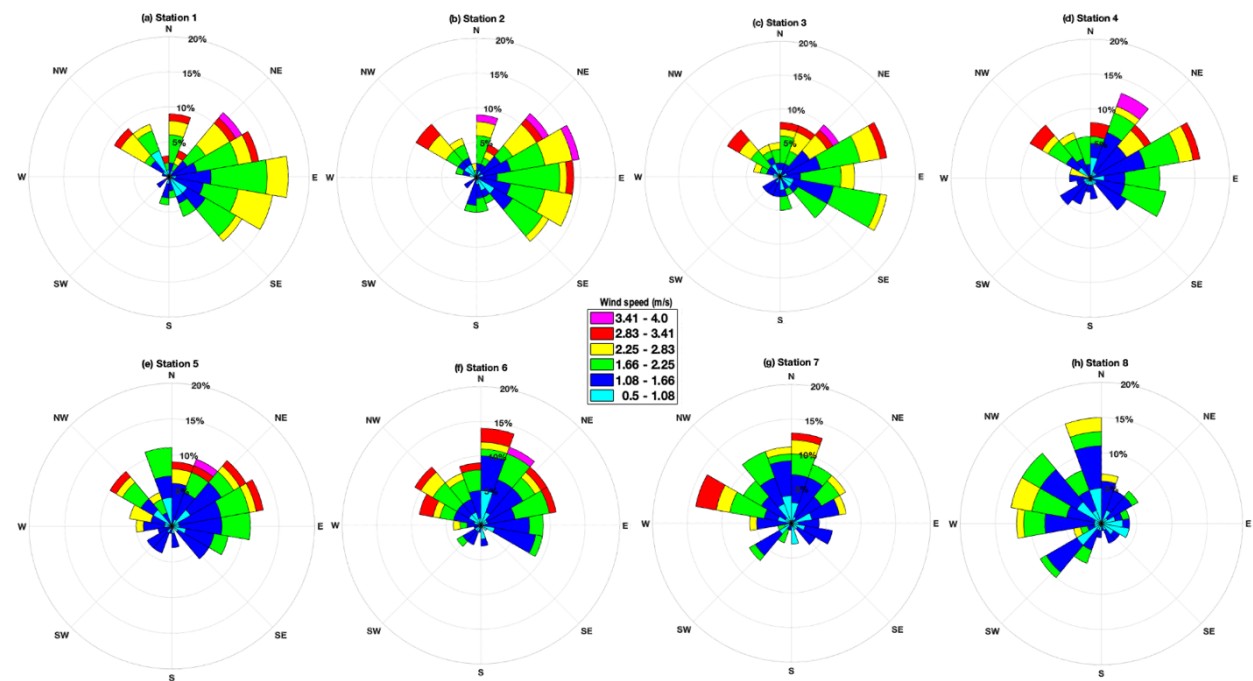

**Figure 5.** Wind rose diagrams at the control points shown in Fig. 1 based on monthly average winds throughout 2010-2019. The wind rose shows the direction the winds come from *[N: North, NE: Northeast, E: East, SE: Southeast, S: South, SW: Southwest, W: West, NW: Northwest].*

To better study the wind characteristics along the ER coast, the wind rose diagrams are shown for the eight control points in Figs. 5(a) to (h). Points 1 to 5, belonging to Zone A and Zone B have the highest wind speeds approaching at an angle 45° to 135°. The wind speed ranging from 3 to 4 m/s is more frequent at these control points at an approaching angle ranging from 45° to 90°. The average coastal angles of Zone A and Zone B are nearly 45°. The points 6 to 8 fall along the concave side of the coastal area i.e., in Zone C. Along these control points, the maximum wind speed approaches from W to NNW. The wind speeds up to 3.5 m/s show a marked increase in frequency. The frequent wind speeds are approaching from NW, and ENE for station 6, NNE for point 7, and NNW for point 8. Moving from point 1 to 8, there is a gentle shift in the maximum wind speed approaching from NNE to ENE.

## 4.2 Wave climatology of the Emilia-Romagna coast

### 4.2.1. Wave height and direction climatology

Figure 6 (top panel) describes the annual mean Hs for the ER coast, and the seasonal Hs means for winter, spring, summer, and autumn. The SD for each event is illustrated in the bottom panels from 2010 to 2019 years. The waves converge at the southern and the northern part of the study domain due to the shape of the coastline. There is divergence in wave energy in the middle region of the coastal domain (i.e., Zone B as reported in Fig. 1c). The annual Hs mean (Fig. 6e) in the domain varied

from 0.08-0.6 m. The annual average Hs is higher (0.5-0.7 m) off the ER coast and at the boundary in the open ocean, and in the central ER domain Hs is of the order 0.5-0.6 m. However, in the ER coastal belt, the annual mean Hs is < 0.4 m owing to the bathymetric features.

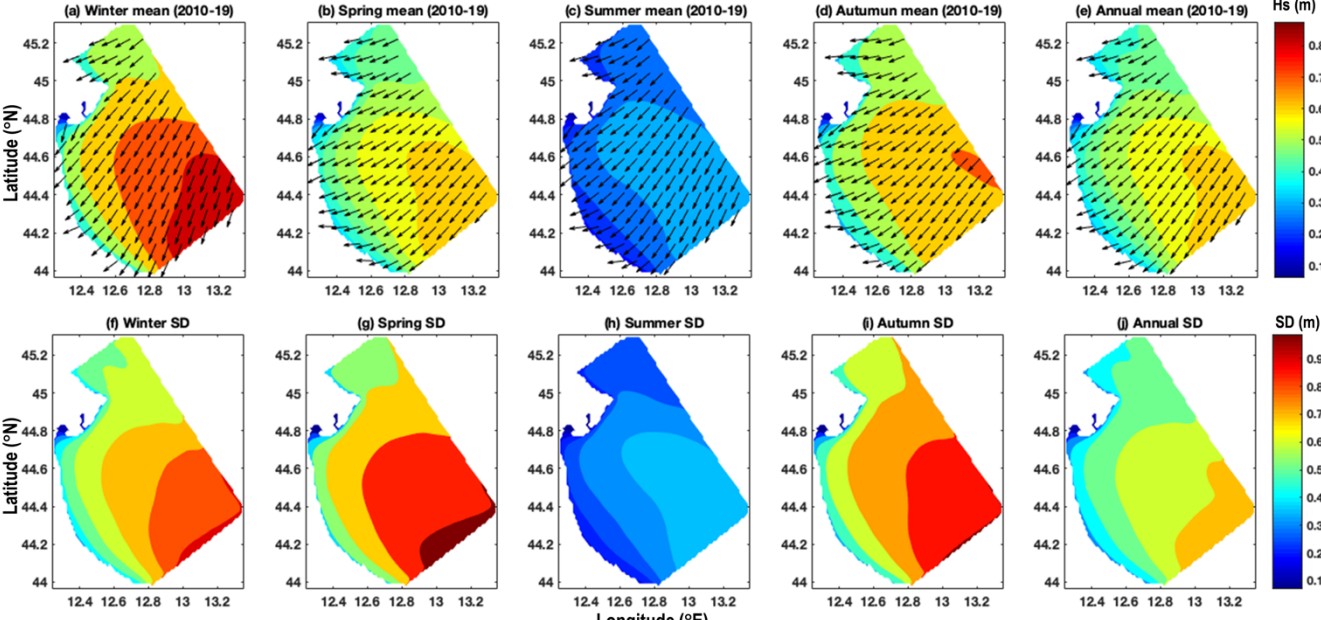

**Figure 6.** Wave climatology for the Emilia-Romagna region for 2010 to 2019. Mean significant wave height and direction for (a) winter, (b) spring, (c) summer, (d) autumn, and (e) annual (top panel). The lower panel shows the standard deviation (SD) of wave height for (f) winter, (g) spring, (h) summer, (i) autumn, and (j) annual periods.

The seasonal climatology of Hs in the winter season (Fig. 6a) indicates higher waves offshore of the order 0.1-0.9 m, where the ER coastal belt has Hs < 0.5 m. In spring (Fig. 6b) and summer (Fig. 6c) the Hs are comparatively lower, and varied in the range of 0.1-0.59 m, and 0.1-0.33 m respectively. The autumn Hs mean in the ER coastal belt is < 0.4 m. The spatial Hs field structure and direction approximately resemble the bathymetric contour lines (Fig. 1b). The annual SD (Fig. 6j) varied from 0.09-0.71 m in the ER domain. The summer season (Fig. 6h) showed the lowest SD (0.1-0.38 m) compared to all other seasons.

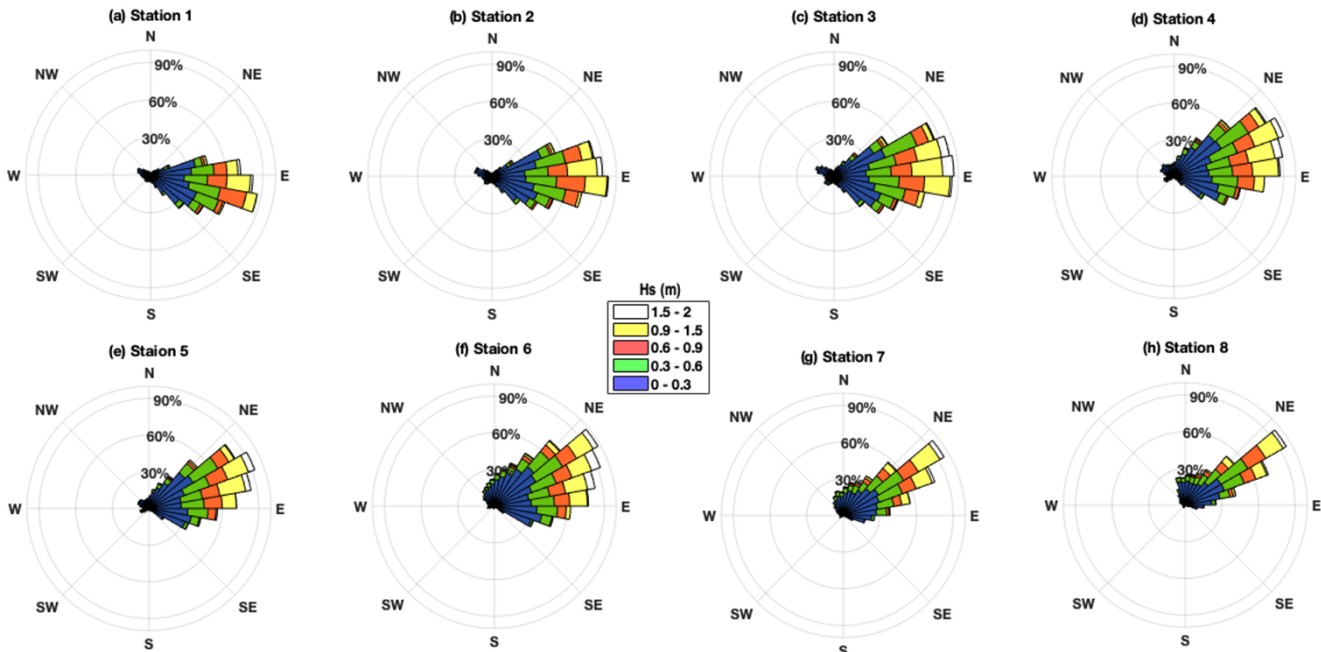

**Figure 7.** Nearshore wave climate: Wave rose diagrams in the coastal belt of Emilia-Romagna along control points 1 to 8 (see Fig. 1c). The wave rose indicates the direction the waves come from *[N: North, NE: Northeast, E: East, SE: Southeast, S: South, SW: Southwest, W: West, NW: Northwest].*

The detailed features of the model in the coastal zone are shown by means of wave rose diagrams (Fig. 7) for the eight points in Fig. 1c. The waves at control point 1 fall in the Lido di Pomposa region where, the coast is sheltered and exposed to winds, and marine currents. The bathymetric contour enables the waves to converge in control point 1, where the maximum wave heights approach from E to SE. From points 2 to 7 along Porto Garibaldi to Rimini, the approaching angles of wave heights are from NE to SE. The maximum waves approach from ENE to E for points 2 to 4, and NE to E for points 5 to 8. The maximum wave activity is observed at point 3. Point 1 is a relatively calmer area compared to the other control points, perhaps because of the shadow zone. The concave shape of the coast, well represented by the high-resolution unstructured-grid model, and bathymetric patterns are key to understanding the prevailing wave characteristics in the ER coastal belt. The wave energy converges at the end points and diverges at the middle points.

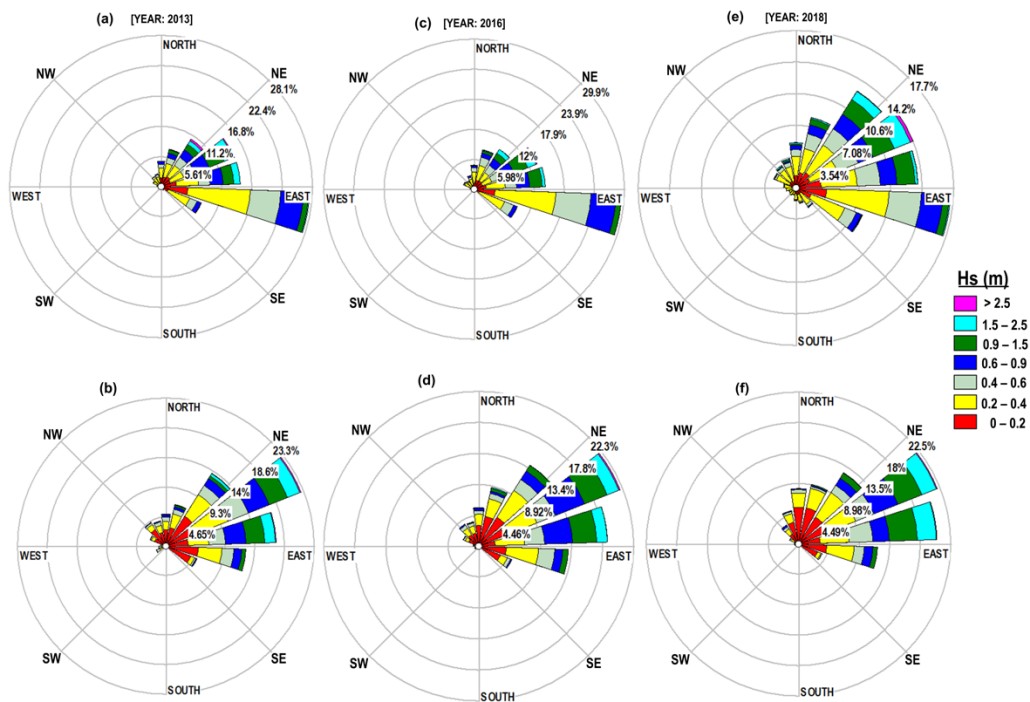

**Figure 8.** Comparison of directional histograms of wave heights: buoy **(a, c, e)**, and simulated **(b, d, f)** data at Cesenatico, station 6 *[NE: Northeast, SE: Southeast, SW: Southwest, NW: Northwest].*

Based on available buoy data for the Cesenatico station the observed wave roses are compared with the model estimates for selected years as shown in Fig. 8. Overall, the modelled wave roses (Figs. 8b, d, f) shows a reasonable correspondence with the observed data (Figs. 8a, c, e), even with some difference in magnitudes. An underestimation of model wave heights in the lower ranges is noted. Comparing the directional distributional of waves, the directions are comparable being in the same sectors but there exist higher differences in their magnitudes. Similar wave climate by the Nausicaa buoy located offshore of Cesenatico is reported in studies by Armaroli et al. (2012), and Romagnoli et al. (2021), which shows that this is the representative wave climate of the Emilia-Romagna coast. This qualitative comparison shows that at the Cesenatico station overall characteristics of waves are fairly reproduced by the model.

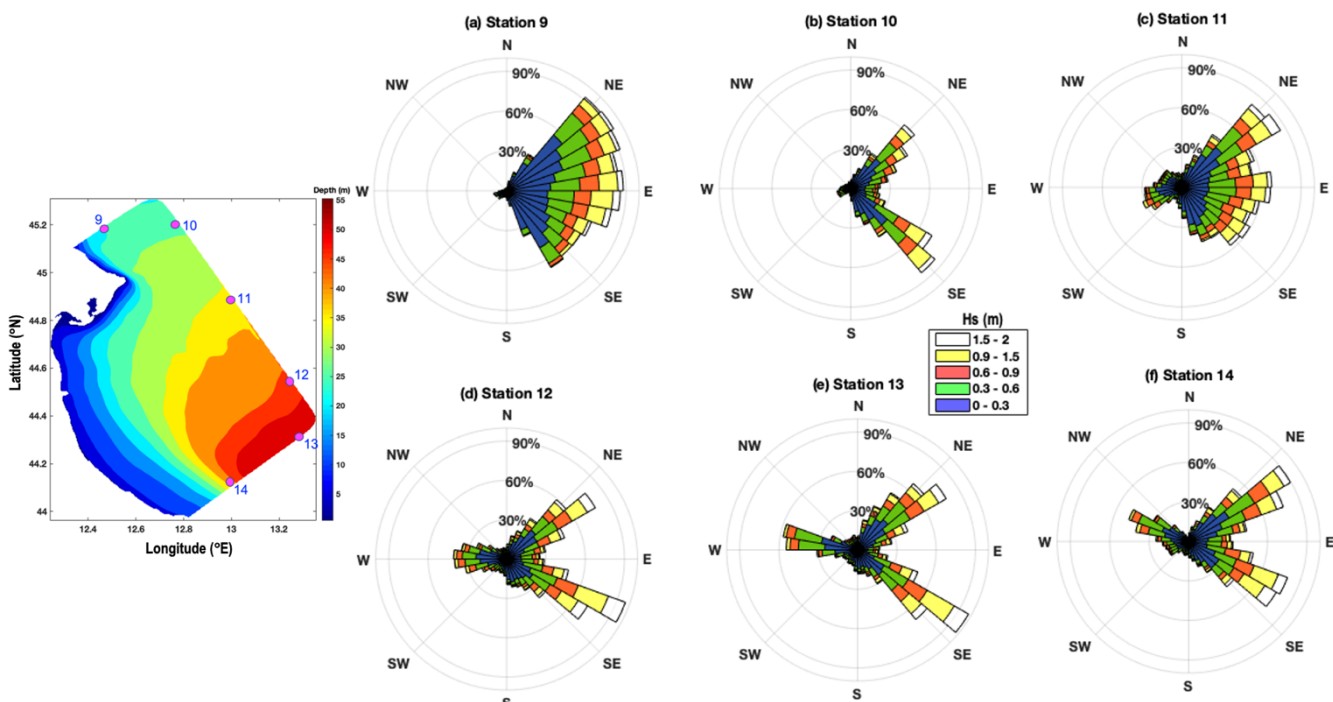

**Figure 9.** Offshore wave climate: Wave rose diagrams in the boundaries of the model domain for the control points (9 to 14) as indicated in the location bathymetric map shown adjacent (left) *[N: North, NE: Northeast, E: East, SE: Southeast, S: South, SW: Southwest, W: West, NW: Northwest].*

Figure 9 reports the offshore wave climate, presented as wave rose diagrams at the control points along the boundaries of the study domains (control points 9 to 14). In Fig. 9(a) and at point 9, the waves approach from NE to SE with maximum Hs approaching from ENE to ESE. At point 10, the predominant waves are at an angle of 30° to 150° where the maximum Hs approach from NE and SE directions (see Fig. 9(b)). For points 11 to 14, the predominant wave directions are from 30° to 150°, where the maximum Hs approach from NE and SE directions. Deep water control points 10 to 14 receive waves from all directions.

### 4.2.2. Wave spectra characteristics

In the ER region, there are hardly any studies on the spectral characteristics of the waves. Cavaleri et al. (2019) analysed a model spectra for the event of October 29, 2018 in Northern Adriatic Sea and compared it with measurements on the Venice coastline. The simulated wave spectra on the 25th of the months corresponding to winter (February), spring (May), summer (August), and autumn (November) at 12:00 hrs are represented in Figs. 10(a) to (d) for station 6 for 2010-2019.

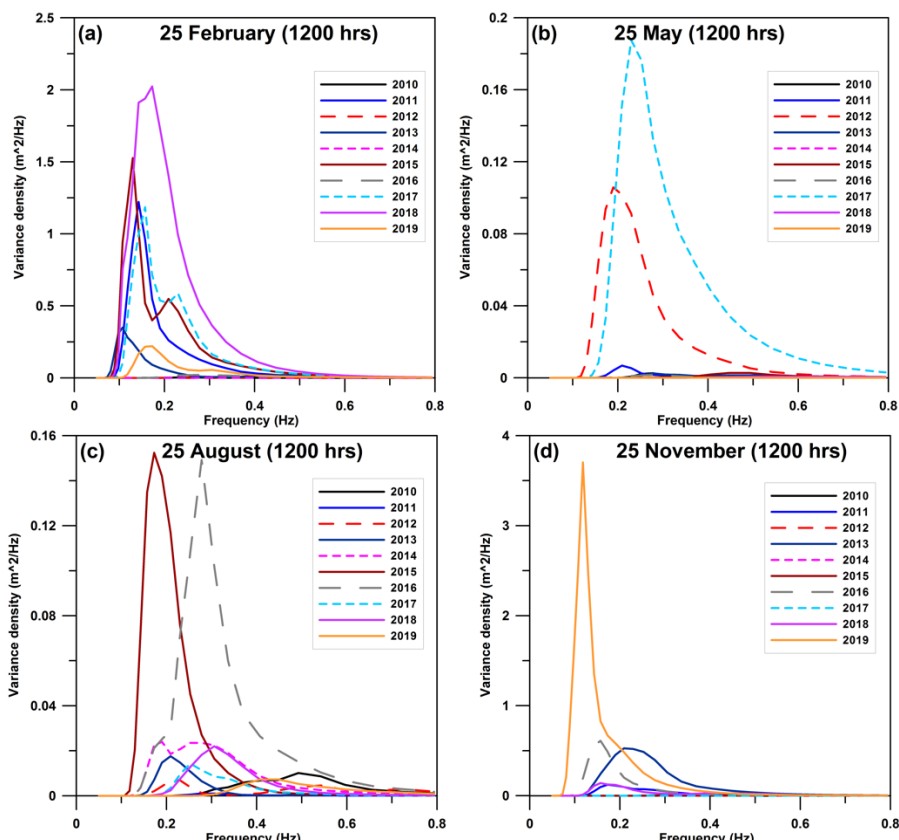

**Figure 10.** Simulated wave spectra for 2010-19 on the 25$^{th}$ day (1200 hrs) of (a) February [winter], (b) May [spring], (c) August [summer], and (d) November (autumn) at station 6 (Cesenatico, see Fig. 1c for location).

Figure 10(a) shows the simulated instantaneous spectra in February (25$^{th}$, 12:00 hrs) with the highest peak energy of 2.0234 m$^2$/Hz for 2018 and the lowest of 0.0008 m$^2$/Hz for 2012 and 2014. February, which is representative month of the winter season, shows a combination of single-peaked and double-peaked spectra with swell dominance at the coastal location. In all the seasons, the swell dominates the spectral energy with a peak at around 0.11 Hz (9 seconds). The shorter wave peaks range

from 0.21 to 0.54 Hz (1.8 to 4.7 seconds). The spectra vary considerably over the years and in general, the wave spectra at the Cesenatico coastal location showed signatures of single and double-peaked spectra for the period 2010-19 (Table 5). The wave spectra were prominently double-peaked during all seasons (45-53 %), along with single-peaked spectra but with a lesser percentage of occurrences (27-33 %). Double peakedness was highly prominent in summer season (53%), while winter, spring and autumn showed dominance of single-peaked spectra (31-33 %). As evident from Table 5, the percentages of the number

of peaks (single/ double) in the Cesenatico location clearly depicts the co-existence of sea-swell characteristics in the study domain.

**Table 5.** Number of occurrences of single-peaked, double-peaked, and multi-peaked spectra at Cesenatico location for different seasons (2010-19).

| Seasons (2010-19) | Single-peak (%) | Double-peak (%) | Multi-peak (%) |
|---|---|---|---|
| Winter | 31 | 45 | 24 |
| Spring | 32 | 45 | 23 |
| Summer | 27 | 53 | 20 |
| Autumn | 33 | 49 | 18 |

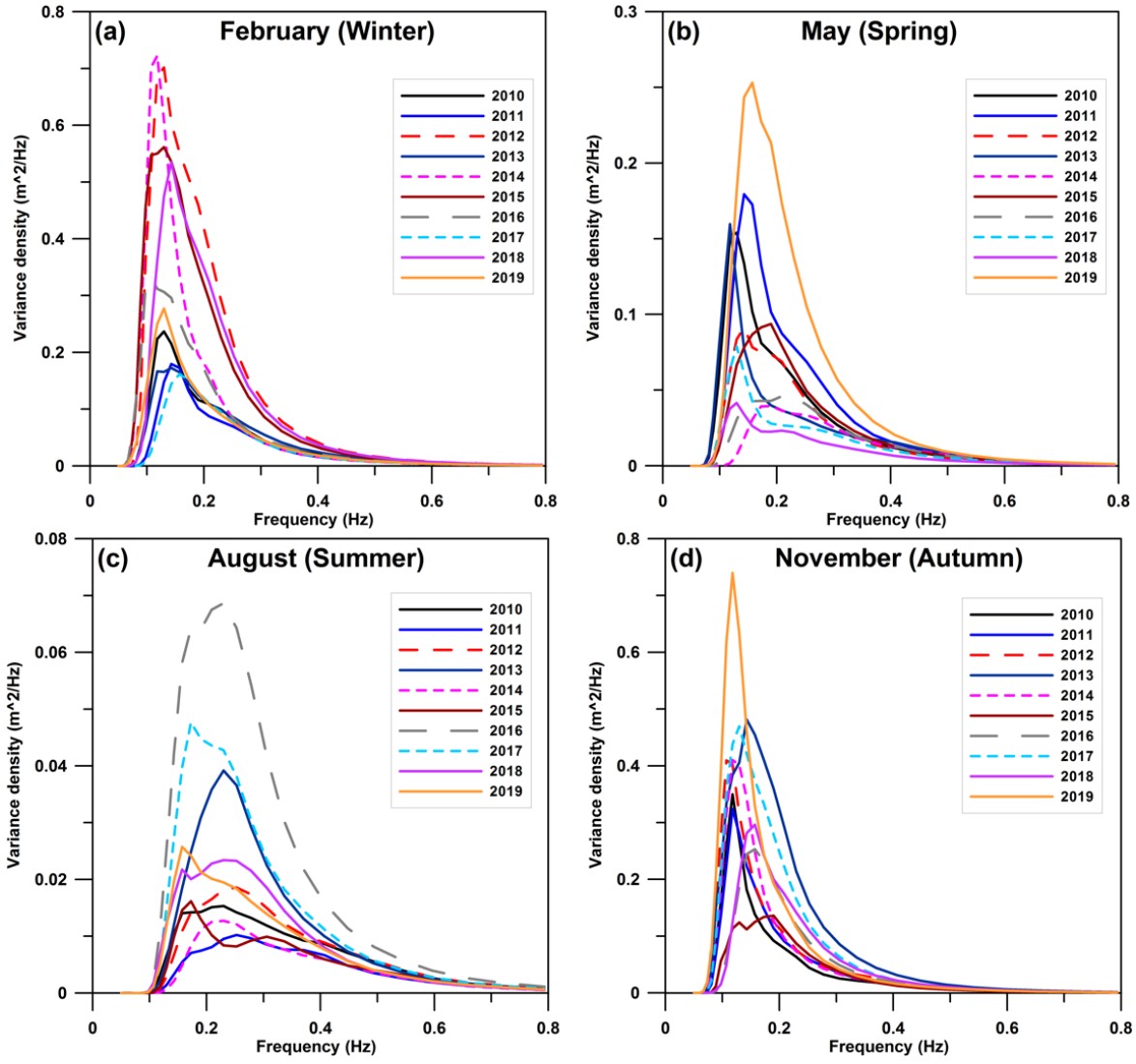

**Figure 11.** Simulated monthly mean wave spectra for the time slice 2010-19 for (a) February [winter], (b) May [spring], (c) August [summer], and (d) November (autumn) at station 6 (Cesenatico, see Fig. 1c for location).

The monthly mean wave spectra for winter, spring, summer, and autumn corresponding to the typical months of February, May, August, and November for 2010-19 are represented in Figs. 11(a) to (d). During February (Fig. 11a), the averaged spectra showed prominent single-peaks for most of the years with peak energies of the order 0.1615-0.722 m$^2$/Hz. The highest peak energies were in 2012 (0.701 m$^2$/Hz), and 2014 (0.722 m$^2$/Hz), and during the 10-year period the peak frequencies ranged from 0.0974 to 0.1726 Hz. Figure 11(b) shows the averaged spectral characteristics for May (spring). As seen from the Fig. 11(b), 2019 had the highest peak energies of 0.253 m$^2$/Hz, and the spectra also highlights a few secondary peaks in some of the years with the peak frequency ranging from 0.1072 to 0.2297 Hz. During the summer season (August), the spectra show single/ double-peaks with peak energies varying from 0.0102 to 0.0686 m$^2$/Hz (Fig. 11(c)). The maximum peak energy was for 2016 (0.0686 m$^2$/Hz) with comparatively lesser energies for the rest of the years, with peak frequencies varying from 0.1427 to 0.278 Hz. Similarly, during autumn (Fig. 11(d)), the averaged spectra was mostly single-peaked with peak energies of the order 0.1362 to 0.740 m$^2$/Hz. The highest peaks with energies of 0.740 m$^2$/Hz were in 2019, with the lowest energy in 2015. The dominant frequencies corresponding to the peak energies were 0.0974-0.2089 Hz.

Overall, the highest and lowest spectral peaks are in winter and summer, with energies of 0.722 and 0.0686 m$^2$/Hz, as shown in Figs. 11(a) and (c). The mean wave spectra for 2010 to 2019 exhibits a peak in variance for 2014, 2019, 2016, and 2019 for winter, spring, summer, and autumn, respectively. The spectra show more or less similar characteristics for spring and autumn. There is also a reversal of spectrum curves for winter and spring, as swells clearly dominate the coastal location. The spreading of spectra is variable during all seasons which is dependent upon the wind characteristics, and the prevailing fetch.

### 4.2.3. Wavelet Analysis

Wavelet analysis is an important tool to analyse spectral components, and the occurrence time (Torrence and Compo, 1998). The wavelet considers spectral components time localization, and time –frequency rendering of signal into realization, such that the frequencies in the wavelet analysis are associated with the time domain. Thus, wavelet analysis (based on, Morlet mother-wavelet) provides an understanding of spectral characteristics, and its variability in time.

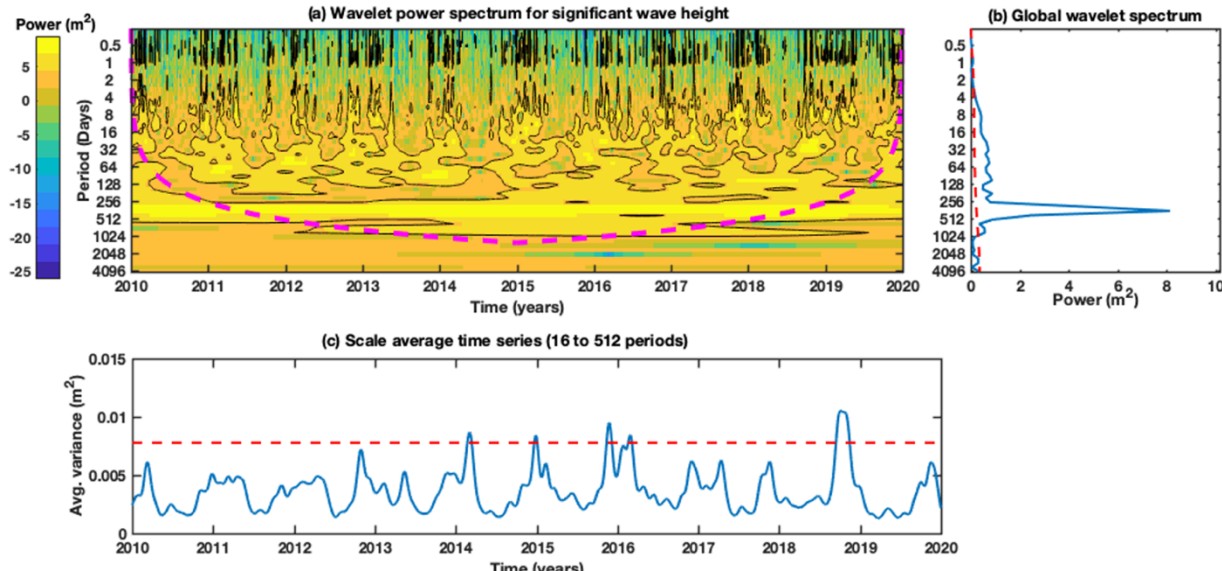

**Figure 12.** Wavelet analysis of wave climate time series (Hs in m) along the Emilia-Romagna coastal belt at station 6 (Cesenatico, see Fig. 1c for location) using mean model estimates (a) wavelet power spectrum for Hs. The colour bar stands for the formation of Hs variation. Power spectra intensity is represented by colours varying from navy blue colour (i.e., weak) to dark yellow (i.e., strong). The contours represent the total variance percentage, and the black contours indicates amplitude significance (greater than 95% level). The dashed magenta line is the cone of influence (region of spectrum with the significant edge effects), where zero padding has reduced the variance. Fig. 11a shows that power is concentrated in the 256-512-days band which is a strong signal, (b) global wavelet power spectrum, where the blue curve indicates the Fast Fourier Transform of the complete data. The dashed red line is the significance (95%) for the global wavelet spectrum, assuming the same significance level and background spectrum as in wavelet power spectra, and (c) scaled-averaged time series over a 16-512-days band showing variance of Hs. The red dashed line is the 95% confidence level for Hs.

In this study the wavelet transform for Hs (Fig. 12) was applied to the coastal location of Cesenatico for 2010-19, using the mean model estimates. Fig. 12(a) represents the wavelet power, with the X-axis representing the time, and Y-axis denoting the component periods. Fig. 12(b) represents the global wavelet power spectrum, i.e., time -averaged power spectrum, which uses the same Y-axis. Cesenatico was selected as it was the station where the wave parameters were validated with the model estimates. The idea of presenting the wavelet transform is to accurately represent the variance in the spectrum. In the power wavelet (Fig. 12a), the real signals can be observed enclosed in the black contours with a 95% confidence level, while the region below the dashed magenta line indicates the cone of influence, in which the time-series analysis edge effects are significant. In the global spectrum, the peaks indicate the combined signal throughout the analysis. The dashed red line in the global spectrum corresponds to a confidence level of 95%.

In Fig. 12, the largest signal occurs in the 256–512-day band which contains the seasonal frequency and sporadic signals can be identified by comparatively shorter times (2-3 months). Fig. 12a indicates that over the 10-year period, intermittent oscillations are in the band 16-128 in the years 2011, 2012, 2014, 2016, 2018, and 2019. Fig. 12c shows the 16–512-day period of the scale average Hs time series, with 95% significance denoted by a dotted red line. Significant peaks can be seen in 2014,

2015, 2016, and 2019 while 2019 shows the highest variance. From 2010 to 2013, and 2017 to 2018 the peaks showed lower amplitudes. The seasonal signal is very different from year to year with peaks occurring sometimes only during the autumn.

**5. Extreme Wave Analysis**

In this study, the statistical characteristics of Hs were analyzed using the methodology of fitting a probability distribution functions (PDF) to the wave time series at the control points of the ER coastal belt. Many studies have indicated that the

470 probability distribution used to model long-term distributions of wave heights are well represented by the two-parameter Weibull distribution (Muraleedharan et al., 1993, 1998, 1999). The PDF of a random variable x with the Weibull distribution (Weibull, 1951) is defined for positive values, x>0, as:

$$f_W(x; \lambda, \kappa) = \frac{\kappa}{\lambda}\left(\frac{x}{\lambda}\right)^{k-1} \exp\left[-\left(\frac{x}{\lambda}\right)^k\right] \qquad (5)$$

where $\kappa$, $\lambda$ ($>0$) are the shape parameter (dimensionless) and scale parameter (m), respectively. It is clear that when $\kappa = 1$, the PDF reduces to an exponential distribution. Fitting this PDF to the data, enables the hazards index to be calculated, which is the probability that the waves will exceed a threshold, let's say $x_c$ in Hs. The hazard index is then defined as:

$$H(x_c) = e^{-\left(\frac{x_c}{\lambda}\right)^k} \qquad (6)$$

To compute the best-fit shape and scale parameters of the Weibull distribution for each of the eight control points, the maximum likelihood method (MLM) was used. This is the most widely used technique among parameter estimations which finds a value of the parameter that maximizes the likelihood function. The values of the Weibull parameters for each control points are presented in Table 6, which shows that the mean, standard deviation, and skewness computed from the model data are very similar to those estimated from the Weibull fit parameters. This thus highlights that the Weibull distribution well

represents the behavior of the Hs model data. The mean value and the corresponding variance of Hs at the Cesenatico station are larger than the other control points, as the station is far from the coast, with the highest water depth. The analysis results show that the fitted Weibull distributions have positive kurtosis, which indicates that the distribution has fat tails.

**Table 6.** The best-fit Weibull scale and shape parameters for Hs (columns 3-4) at the eight control points. Column 5 shows the estimated $\chi^2$ values. Mean, variance, skewness, and kurtosis of the Hs (columns 6-9) computed from the model data (left sub-columns) and from the Weibull fit parameters (right sub-columns), and the wave height hazard index (with threshold value Xc, Hs=1.08m) calculated with (eq. 6) for the eight control points (indicated in column 10) along the Emilia-Romagna coastal strip.

| Control points | Station Name | Scale $\lambda$ | Shape $\kappa$ | chi$^2$ | Mean (m) | | Variance (m$^2$) | | Skewness | | Kurtosis | | Hazard Index |
|---|---|---|---|---|---|---|---|---|---|---|---|---|---|
| | | | | | Model | Estim. | Model | Estim. | Model | Estim. | Model | Estim. | |
| 1. | Lido di Pomposa | 0.32 | 1.01 | 0.42 | 0.32 | 0.32 | 0.108 | 0.101 | 1.69 | 1.97 | 3.01 | 5.79 | 0.01 |
| 2. | Porto Garibaldi | 0.35 | 1.03 | 0.44 | 0.35 | 0.35 | 0.124 | 0.115 | 1.64 | 1.92 | 2.46 | 5.47 | 0.04 |
| 3. | Casalborsetti | 0.39 | 1.06 | 0.56 | 0.38 | 0.38 | 0.142 | 0.133 | 1.56 | 1.84 | 2.03 | 5.01 | 0.05 |
| 4. | Lido Adriano | 0.39 | 1.00 | 0.27 | 0.39 | 0.39 | 0.167 | 0.151 | 1.88 | 2.00 | 4.09 | 6.00 | 0.06 |
| 5. | Foce Savio | 0.35 | 1.00 | 0.48 | 0.35 | 0.35 | 0.133 | 0.125 | 1.68 | 2.00 | 2.67 | 6.02 | 0.05 |
| 6. | Cesenatico | 0.39 | 0.95 | 0.25 | 0.41 | 0.40 | 0.210 | 0.183 | 2.11 | 2.18 | 5.44 | 7.23 | 0.07 |
| 7. | Rimini | 0.36 | 0.96 | 0.35 | 0.37 | 0.37 | 0.169 | 0.147 | 2.16 | 2.13 | 5.91 | 6.88 | 0.06 |
| 8. | Riccione | 0.34 | 1.00 | 0.56 | 0.34 | 0.34 | 0.129 | 0.116 | 1.97 | 2.00 | 4.72 | 6.02 | 0.04 |

To evaluate the goodness-of-fit of the Weibull distribution, the classical chi-square ($\chi^2$) test was used. This test determines how well the theoretical distribution fits the given model data distribution. If the chi-square value is lower than a critical $\chi^2$ value, we retain the null hypothesis, and conclude that there is no significant difference between the observed and the expected distributions. The estimated $\chi^2$ values for each control points are given in Table 6. The decision rule for the $\chi^2$ test depends on the level of significance (set to 0.05) and the degrees of freedom, defined as df=N-np (where N is the number of bins (set to 30), and np is the number of distribution parameters (i.e., 2)), so that the critical value of $\chi^2$ is 41.34 (taken from the $\chi^2$ distribution table). Table 6 highlights that the two-parameter Weibull distribution fits the Hs data well.

In all the eight locations, since the shape value $\kappa$ is close to 1, the fitted Weibull distributions behave like the exponential distribution. Fig. 13 compares the Weibull distributions fit (red line) and the histogram of the model data for three relevant locations (Porto Garibaldi, Lido Adriano, and Cesenatico). The two-parameter Weibull distribution appears to fit the data well in the coastal study area.

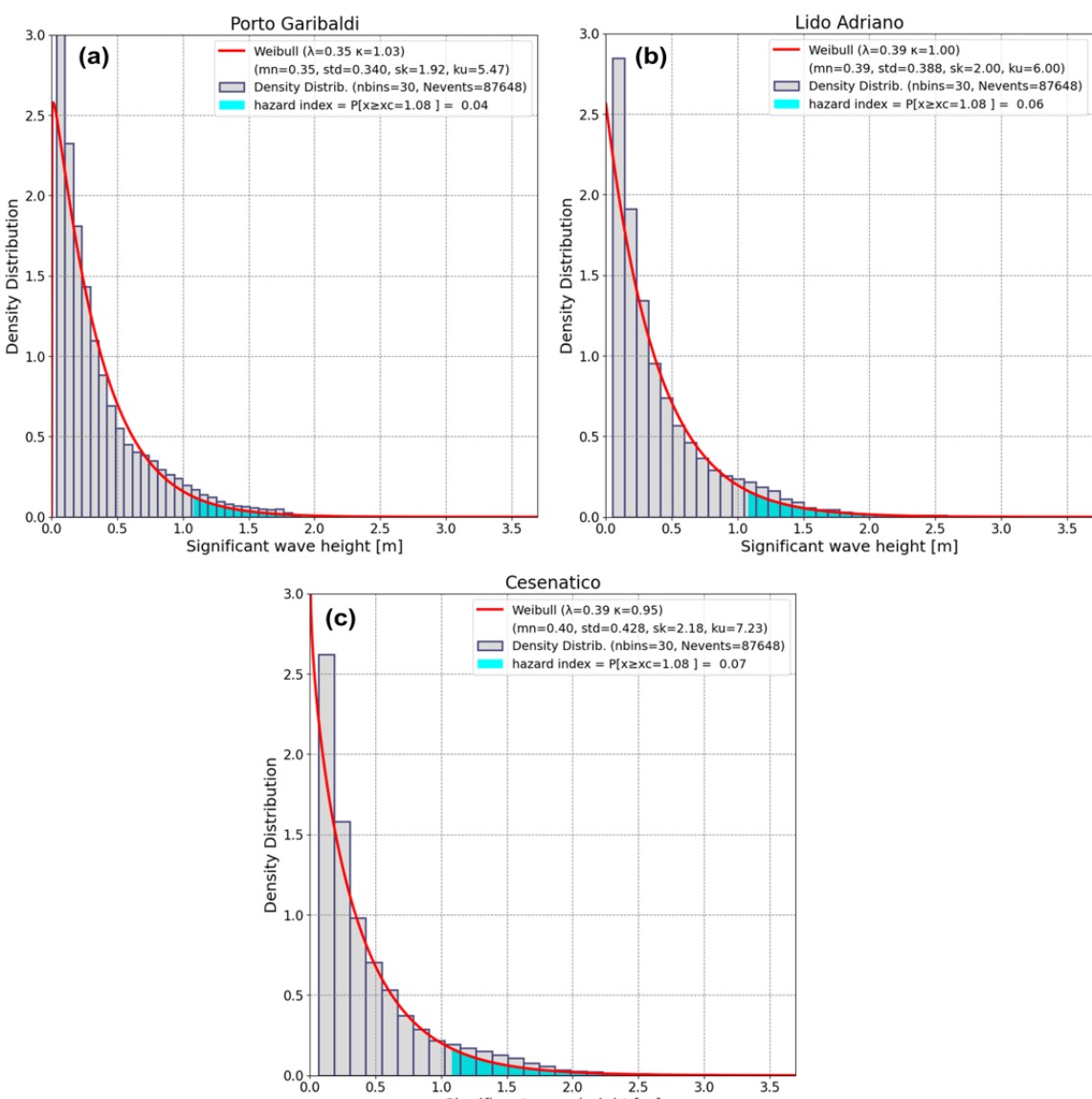

**Figure 13.** Comparison of the Weibull distribution fit (red line) to the histogram of the model data (2010-2019) for the control points: (a) Porto Garibaldi, (b) Lido Adriano, and (c) Cesenatico. The red line denotes the Weibull fit, the histograms represent the density distribution (seen in grey color), and the hazard index is indicated in cyan color *[mn: mean, std: standard deviation, sk: skewness, ku: kurtosis, nbins: number of bins, Nevents: number of events].*

After evaluating the Weibull distribution fit and the statistical moments, we estimated the hazard index as shown in Table 6 (column 10) for a threshold value Xc (i.e., Hs=1.08 m, 3 times the mean standard deviation). The hazards were shown to increases 7-fold from the northern control points (Lido di Pomposa) to Cesenatico and then to decreases again. In the future it will be interesting to compare the hazards for different coastal areas around the Adriatic Sea.

# 6. Summary and conclusions

To accurately simulate the wind-wave climate in the Emilia-Romagna coastal belt, a high-resolution numerical modelling study using unstructured-grid WW3 was executed for a 10-year period. The WW3 model was driven by the ECMWF analysis winds and the model was validated with available wave buoy data at a coastal location. The sensitivity tests has shown the good accuracy of ST6+SHOWEX physics for wave hindcasts in the study area. The results of a comparison of model estimates with measurements were promising. An Hs correlation of 0.86 to 0.93 was found for the 10-year simulations with observed data. The underestimations in Hs were indicative of a negative bias (-0.076 m to -0.016 m) with an RMSE of 0.19 m to 0.25 m. The comparison of Tm and Tp with buoy measurements revealed a correlation of 0.70 to 0.80, and 0.53 to 0.70, respectively, for the 10 years. Our database was used then to study and characterize the present climate of waves for the region, and a hazard index for extreme events was defined and computed. The following conclusions were drawn:

- The spatial mean wind speed for winter, spring, summer, autumn varied in the range 1.1-2.9 m/s, 0.5-1.5 m/s, 0.5-1.8 m/s, and 0.5-2.4 m/s respectively. The lowest wind speeds were observed during spring and summer (1.5/1.8 m/s) considering the study domain, followed by autumn (2.4 m/s) and with the highest wind speeds (2.9 m/s) observed during winter seasons.

- The annual Hs mean in the ER domain varied from 0.08-0.6 m, and in the ER coastal belt the annual mean Hs was < 0.4 m owing to the bathymetric features. In the ER coastal belt, the seasonal climatology of Hs in the winter showed a mean Hs < 0.5 m, while in spring and summer the Hs were comparatively lower (Hs < 0.38 m/ Hs < 0.21m). The autumn Hs mean is < 0.4 m. It should be noted that there was more wave activity in winter and autumn than in spring and summer.

- The analysis of instantaneous spectra showed that during all seasons the spectra exhibited bi-modal characteristics (double-peaked), with a dominant occurrence percentage of 53 % in summer, while during spring, winter, and autumn the spectra also showed single-peaked spectra (31-33 %). The average spectra analysis showed peak frequencies of the order 0.097 to 0.172 Hz, 0.107 to 0.229 Hz, 0.142 to 0.278 Hz, and 0.097 to 0.208 Hz for winter, spring, summer, and autumn seasons respectively.

- With the aid of a wavelet analysis tool, the power features (time -frequency) of Hs data showed substantial variability of Hs for a 10-year period, with the occurrence of monthly and seasonal periods. The 256–512-day band showed a higher power concentration which represents the seasonal frequency.

- The coastal control points time series was well fitted by a Weibull PDF. The Weibull at all control points was congruent with an exponential distribution. Using the Weibull PDF fit, we calculated a hazard index which indicated that for waves higher than 3 standard deviations from the mean, i.e., the highest hazard reached at the Cesenatico station.

The limitation in the present study is the non-availability of wave spectra measurements at the coastal locations for validation. Future study will aim to consider data assimilation (de Rosnay et al., 2022) and to have higher resolutions winds as forcings for the wave model. In the context of heavy tailed data sets the Weibull distribution may not represent a best description of the

peak and tail. These limits can be overcome by adding more parameters such as the 4-parameters exponentiated Weibull distribution (Mahmoudi et al., 2018) such that the extra shape parameter can provide more versatility to the distribution in the shape of the tails.

Another limitation is that a 10-year period would generally not be enough to bring out the complete climatological wave response to winds. ECMWF winds were too low resolution before 2010 and no downscaled limited area meteorological forcing is available. Hence this 10-year period could be the first reference database for the prevailing wind-wave characteristics in the coastal belt for researchers and coastal engineers/designers. Future works definitely should deal with longer simulation periods and also higher resolution winds.

Our analysis highlights the importance of long-term wave databases, which can aid in the design requirements of coastal engineering applications. It also demonstrates the useful application of PDF to the estimate of hazards along the coastal belts. The study also highlights the need for extensive wave spectra comparisons (Lobeto et al., 2021) with measurements for selected locations on the coastal belt which will update the coastal wave database. The early detection of hazards such as coastal erosion, and associated shoreline changes are still demanding (Le Cozannet, et al., 2020), due to the non-availability of long-term observations. As reported by Vousdoukas et al. (2018), by the end of the century, the community encountering marine flooding 580 is estimated to rise from 1.52 to 3.65 million, and considering the global vulnerability (Luijendijk et al., 2018), low lying nearshore regions (one-fourth) are retreating, and the eroded land (Mentaschi et al., 2018) remains as twice what is acquired. Better knowledge of the prevailing wave characteristics on the ER coastal belt will aid in predicting the coastal impacts.

**Code and data availability.** The data/codes used in this study can be accessed at the Zenodo archive: https://doi.org/10.5281/zenodo.6360348.

**Author contributions. UPA:** Conceptualization, data curation, investigation/ analysis, methodology, validation, and visualization, results interpretation, writing the original draft; **NP:** project supervision, conceptualization, methodology, results 590 interpretation, writing - review & editing; **IF:** data curation, methodology, writing - review & editing; **SC:** data curation, methodology, writing - review & editing; **FT:** data curation, methodology, results interpretation, writing - review & editing; **SU:** data curation, writing - review & editing**; AV:** data curation, writing - review & editing.

**Competing interests.** The authors declare that they have no conflict of interest.

**Acknowledgements.** This work was carried out under the framework of OPERANDUM (OPEn-air laboRAtories for Nature baseD solUtions to Manage hydro- meteo risks) project, which is funded by the European Union's Horizon 2020 research and innovation programme under the Grant Agreement No: 776848.

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
