# Peer review of "Wind-Wave Characteristics and extremes along the Emilia-Romagna coast"

_Natural Hazards and Earth System Sciences, 2022_

## Referee Comment (RC1)

**Manuscript ID: nhess-2022-103**

**General comments**

The manuscript (MS) examine an in-depth analysis of wind-wave characteristics along Emilia-Romagna (ER) coasts (northern Adriatic Sea, Italy) for 10 years starting from 2010 after fine tuning the numerical model-Wave Watch III at a buoy location. I appreciate the author's efforts to include multidimensional analysis for better understanding of the seasonal and extreme wave impacts along the selected coast. The flow of the MS and specially discussion of research gap gives a clear idea for a reader about this work. There is no doubt about the quality of work. But these are some suggestions for authors which can even elevate the clarity of this research.

  a) There is a great discussion on wind speed in the MS. The wind speed values (section 4.1.1) seems like a low range for me. I strongly recommend the authors to check those values with in-situ measurements if you have any.
  b) Authors could discuss a bit more literatures on research conducted in similar way along global study regions than Adriatic Sea
  c) Could 10 years include the climatological impacts of wind-wave characteristics? The MS analyse more of seasonal aspects than climatological/long-term variation.

Some specific comments are mentioned below

**Specific comments**

Kindly go through the below mentioned comments and alter the MS wherever is necessary:

1. It will be greatly appreciated to mark the names mentioned in text to be on the figures. For eg. Line 74 mentions about Po delta, it will be easier for the readers to understand the work more if it's marked in the figure too for visualization.

2. Line 26: Than IPCC,2007, now IPCC report citations can be included.

3. What is restricted/controlled fetch? It will be good to get more clarity on MS too.

4. Line 85: Indicating the dominance of swell or sea in the selected study domain can enhance the knowledge beforehand.

5. Figure 1: This can be more legible and well distributed. The mesh and bathymetry info can be in one figure. The figure is not indicating anything like region, the Sea etc. Including that can be a good idea. Figure 1. (a) can be an inset image and other information are important could be enlarged. A legible north arrow with map scales, coordinates degree N, E etc are also recommended (this comment is applicable for all maps).

6. Give the legend names (eg Figure 7, wave height (m) than mentioning meter in legend ranges), x- and y-axis variables in each plots, etc

7.  Line 199: Mention which ECMWF wind: ERA5, ERA interim etc.

8.  Appreciate explaining why the zones or control points are assigned? Or on what basis?

9.  Line 156: Why there is no ST6+ JONSWAP (EXP4)?

10. Line 157: Apt to mention why the representative months are February and September?

11. Section 3.3: Why the validation was done for each year separately? What is the significance of that?

12. Figure 3 (f-j): It might be suitable to use 45-degree line than best fit line to best understand the underestimation and bias in validation. Check for wave characteristic notations in graph axis too.

13. Sub-section 4.1.1 is the only subsection under 4.1, which can be merged with section 4.1 itself.

14. Section 4.1.1: The wind speed values seems to be lower than a desired value. And SD is more than the annual mean wind speed which is not right. Check those values for wind speed everywhere. The mean value can come around 5 m/s to 20 m/s to produce the wave characteristics indicated in the MS. Make changes in text and figures accordingly. I recommend comparing these values with in-situ measurement if you have any for authors clarity.

15. Figure 4 &6, mark degree N, E on Latitude and Longitude

16. Revise Figure 5 according to new wind speed data and try to correlate Figure 5 and Figure 7 for any influence of predominant wind direction on wave characteristics.

17. Line 285: Sticking to one notation of position can help the reader understand better. Either 'control points' or 'stations. Indicating 'point' can sometimes make confusions.

18. What is the significance of analysis of $25^{th}$ day of month and monthly mean seasonal spectra for each year? What are the concluding remarks of these analysis could be mentioned?

19. Line 315: 'As seen from the Fig.???' to be filled

20. Line 425: 'the comparison of Tm and Tp…… 10 years' needs clarity. Please reframe by adding adequate information.

21. A discussion on limitations/uncertainties of this study could be added. Such as limitations of Weibull distribution,

**Technical /minor corrections**

*Some of the minor technical/typos noticed are mentioned here:*

i.    Line 13: 'direction' to 'wave direction'

ii. Significant wave height could be H subscript s. This applies with every wave characteristic. Using the global notations can be beneficial for a wider audience in understanding this research more.

iii. Expand acronym at first appearance will be appreciated. Few eg: are line 26 IPCC, WW3 is not expanded anywhere in MS, Line 101: 'JONSWAP parameterization (Joint North Sea Wave Project)', this can be 'Joint North Sea Wave Project (JONSWAP) parameterization', Explain ST4, ST6, CFL etc. Could use the acronyms after defining. PDF is expanded twice and is using the same.

iv. Line 121: check '~4.5 km hourly'

v. After defining Significant wave height ($H_S$) use the same in everywhere.

vi. I recommend the authors to make the decimal places consistent in entire MS. Eg. Line 126 location coordinates has 4 decimal places. Most research work would go for 2 decimal numbers.

vii. Line 180: 'measurement's' to 'measurements'

viii. Line 188: 'Fig. 3' to 'Figure 3'

ix. Line 190: 'relatively a good' to 'relatively good'

x. Line 250: 'and northern' to 'and the northern'

xi. Line 327: 'upon blowing of the wind' to 'wind characteristics'

xii. Line 348: 'costal' to 'coastal'

xiii. Line 423: 'A Hs' to 'An'

xiv. Line 435: 'waves' to 'wave'

---

## Author Response (AR1)

**Dr. Umesh P. A.**
University of Bologna
ITALY

**The Editor**
Dr. Joanna Staneva
Natural Hazards and Earth System Sciences

**Dear Dr. Joanna Staneva,**

Please find attached the responses to the reviewer comments for the manuscript **NHESS-2022-103** entitled *"Wind-Wave Characteristics and extremes along the Emilia-Romagna coast"*.

We have carefully examined the constructive suggestions made by the reviewers and we have taken full account of their comments. Hence, the necessary corrections and modifications are incorporated as suggested by the reviewers. The following is a point-by-point response to the comments and inquiries made by **reviewers 1 & 2**.

We would like to thank the reviewers for the constructive and competent comments/suggestions which helped to improve the manuscript highly in terms of clarity and readability.

**Respectfully yours,**

**Umesh P. A. (on behalf of all co-authors)**

**Responses to Reviewer 1:**

**Dear Reviewer,**

We thank you for the thorough evaluation of our manuscript. The constructive comments and suggestions have improved the manuscript highly. Below, we address each of your comments in turn. The comment is repeated in italics and our authors' response (AR) directly follows highlighted in blue color. The AR contains our reply and a brief description of what has been modified in the manuscript. In the revised manuscript, the corrections suggested have been addressed.

**Best Regards,**
**Umesh (on behalf of all co-authors)**

**General comments**

*The manuscript (MS) examine an in-depth analysis of wind-wave characteristics along Emilia-Romagna (ER) coasts (northern Adriatic Sea, Italy) for 10 years starting from 2010 after fine tuning the numerical model-Wave Watch III at a buoy location. I appreciate the author's efforts to include multidimensional analysis for better understanding of the seasonal and extreme wave impacts along the selected coast. The flow of the MS and specially discussion of research gap gives a clear idea for a reader about this work. There is no doubt about the quality of work. But these are some suggestions for authors which can even elevate the clarity of this research.*

**AR:** Thank you for your in-depth and detailed evaluation of our manuscript. The authors have noted the reviewer's suggestions and comments. The corrections suggested by reviewer #1 are incorporated into the revised version of our manuscript.

**a)** *There is a great discussion on wind speed in the MS. The wind speed values (section 4.1.1) seems like a low range for me. I strongly recommend the authors to check those values with in-situ measurements if you have any.*

**AR:** The reviewer comment is well appreciated, and a clarification is offered.

The study uses ECMWF analysis winds at 0.125° horizontal resolution every 6 hourly as inputs to the wave model. In section 4.1.1. the Figure 4, presents the **mean wind speed** and direction for the period 2010-2019 in the model domain. As pointed out by the reviewer the wind speed ranges are re-checked and confirmed.
We present a validation of wind speed and direction from available data sets for the locations of Porto Corsini, Porto Garibaldi and Cesenatico Port, locations are given in Table 1.

**Table 1:** Details of coastal stations used for validation of wind forcing.

| Station | Lat (°N) | Lon (°E) |
|---|---|---|
| Porto Corsini | 44.49 | 12.28 |
| Porto Garibaldi | 44.67 | 12.24 |
| Cesenatico Port | 44.20 | 12.40 |

[Figure]

**Figure I:** Comparison of ECMWF winds with measurements at the Porto Corsini station for the year 2013: (a) wind speed (m/s), and (b) wind direction (deg.).

Figure I (a, b) shows the comparison of wind speed and direction with the observations at Porto Corsini for the year 2013. The ECMWF winds show an overall consistency with observations. The comparison statistics for all three stations is shown in Table 2. We see that overall, the correlation at all stations and seasons is higher than 0.7 with the exception of summer and autumn, as expected because of intense air-sea interaction in the coastal boundary layer which are not captured by the ECMWF model. The Bias is low while the root mean square error between observations and ECMWF is about 2 m/s in all seasons. This is visualized also with the wind speed scatter plots in Figure-II.

**Table 2:** Quality assessment of ECMWF winds with observed wind speeds for selected stations (refer Table 1).

| | Wind speed (m/s) | | | | |
|---|---|---|---|---|---|
| **Statistics** | **(a) Porto Corsini [Year: 2013]** | | | | |
| | **Full year** | **Winter** | **Spring** | **Summer** | **Autumn** |
| **R** | 0.7 | 0.7 | 0.7 | 0.5 | 0.7 |
| **Bias** | -0.2 | 0.2 | -0.1 | -0.3 | -0.4 |
| **RMSE** | 1.8 | 1.8 | 1.9 | 1.6 | 2 |
| **(b) Porto Garibaldi [Year: 2018]** | | | | | |
| **R** | 0.7 | 0.8 | 0.7 | 0.5 | 0.8 |
| **Bias** | -0.2 | 0.2 | -0.3 | -0.5 | 0 |
| **RMSE** | 1.8 | 1.7 | 1.6 | 1.9 | 1.9 |
| **(c) Cesenatico Port [Year: 2015]** | | | | | |
| **R** | 0.7 | 0.8 | 0.8 | 0.5 | 0.6 |
| **Bias** | -0.2 | 0 | -0.3 | -0.6 | 0.2 |
| **RMSC** | 1.9 | 1.7 | 2 | 1.9 | 2 |
| *R: Correlation, RMSE: Root Mean Square Error* | | | | | |

[Figure]

**Figure II:** Scatter plot of wind speed (m/s) at the three stations **(a)** Porto Corsini, **(b)** Porto Garibaldi, and **(c)** Cesenatico port. The red dotted lines represent the best data fit and the black dotted lines indicates the 1:1 slope. *[R: Correlation, B: Bias, and RMSE: Root Mean Square Error].*

Thus, we argue that in this region ECMWF wind quality is reasonable. Naturally in the future higher resolution limited area modelling winds could be used to reinforce the characterization of the wind climatology in this area but at this time, long time series are not available.

We added the following explanation at line 140-144 (Page 6 in the revised manuscript), along with Table 1 (line 146, page 6 in the revised manuscript) indicating the wind speed validation statistics in the revised manuscript:

The model winds were validated at three stations, namely Porto Corsini (44.49°N, 12.28°E), Porto Garibaldi (44.67°N, 12.24°E) and Cesenatico Port (44.20°N, 12.40°E) along the ER coastal belt. The wind speed comparison statistics as indicated in Table 1 showed correlations of the order 0.7, with bias of -0.2 m/s indicative of underestimation of wind speed, and RMSE of 1.8 m/s. Larger biases of the order of -0.6 m/s and correlations as low as 0.5 exist during summer and some autumn seasons.

Furthermore, in the conclusions section we added a comment about the limitation of the present study due to the low-resolution winds (**see comment c below**).

**b)** *Authors could discuss a bit more literatures on research conducted in similar way along global study regions than Adriatic Sea*

**AR:** As suggested authors have added a paragraph on similar studies reported across the globe.

The manuscript has been modified to highlight this point as shown below (Page 2, lines: 31-41 in the revised manuscript).

Over the globe, wave climatology studies using reanalysis datasets and model hindcasts are reported by Carter et al. (1991), Sterl et al. (1998), Young (1999), Cox and Swail (2001), Sterl and Caires (2005), Hemer et al. (2010), Semedo et al. (2011), Young et al. (2011), Zheng et al. (2016), and De Leo et al. (2020). Wind speed and wave height climatologies with emphasis on the Southern Ocean is described in the works of Young (1999), Young and Holland (1996), Young and Donelan (2018). Past studies on regional scales (Young et al., 2020) based on observations and numerical modelling were also reported by various researchers on different regions such as: Northern Hemisphere (Woolf et al., 2002; Reistad et al., 2011); Southern Hemisphere (Gorman et al., 2003); Mediterranean Sea (Lionello and Sanna, 2005; Lionello,

2012; Clementi et al., 2017; Ravdas et al., 2018; Morales-Márquez et al., 2020; De Leo et al., 2021; Barbariol et al., 2021; Amarouche et al., 2022), Persian Gulf (Kamranzad et al., 2013), western Australia (Bosserelle et al., 2012), eastern North Atlantic (Dodet et al., 2010), southeast Pacific ocean (Aguirre et al. 2017), Indian Ocean (Stopa and Cheung, 2014), Black Sea (Akpinar and Komurcu, 2013; Arkhipkin et al., 2014; Fedor et al., 2020), and China Seas (Zheng and Li, 2015; Qian et al., 2020).

**c)** *Could 10 years include the climatological impacts of wind-wave characteristics? The MS analyse more of seasonal aspects than climatological/long-term variation.*

**AR:** The reviewer comment is well appreciated, and a clarification is offered:

The aim of the study is to report the wind-wave characteristics specific for the Emilia-Romagna coastal belt. The authors very well agree to the fact that a 10-year period would generally not be enough to bring out the climatological response to winds. However, ECMWF winds were too low resolution before 2010 and no downscaled limited area meteorological forcing is available. Hence this study for a 10-year period could be useful for researchers and coastal engineers/designers as a reference database for the prevailing wind-wave characteristics in the coastal belt. Future works definitely should deal with longer simulation periods and also higher resolution winds.

We added this comment in the conclusions, line 569-573, page 26 in the revised manuscript:

Another limitation is that a 10-year period would generally not be enough to bring out the complete climatological wave response to winds. ECMWF winds were too low resolution before 2010 and no downscaled limited area meteorological forcing is available. Hence this 10-year period could be the first reference database for the prevailing wind-wave characteristics in the coastal belt for researchers and coastal engineers/designers. Future works definitely should deal with longer simulation periods and also higher resolution winds.

**Specific comments**

1. *It will be greatly appreciated to mark the names mentioned in text to be on the figures. For eg. Line 74 mentions about Po delta, it will be easier for the readers to understand the work more if it's marked in the figure too for visualization.*

   **AR:** The reviewer comment is noted and considered. As suggested the Figure 1 (page 4 in the revised manuscript) is corrected by marking the location specific names in the Emilia-Romagna coastal belt for better clarity and visualization.

2. *Line 26: Than IPCC,2007, now IPCC report citations could be included.*

   **AR:** As suggested the more recent IPCC report is added, in lines 27-30, page 1 in the revised manuscript, as shown below:

   IPCC (2021) indicates the necessity of a regional evaluation of climate change, with various target factors that can aid in risk management and policy making. The report points out that over the 21st century nearshore regions will encounter sea level rise, thereby adding to more persistent coastal flooding (across low lying regions) and associated coastal erosion.

**3.** *What is restricted/controlled fetch? It will be good to get more clarity on MS too.*

**AR:** Restricted fetch means a condition where the wave generating area (i.e. fetch) is relatively small. We added the sentence in lines 97-98, page 4.

**4.** *Line 85: Indicating the dominance of swell or sea in the selected study domain can enhance the knowledge beforehand.*

**AR:** As suggested information on sea/swell dominance in the study area is added as shown below:

Thus the swell seas are controlled by the Sirocco winds and the seas are dominated by the Bora winds (Bonaldo et al., 2017).

These changes appear in lines 99-100, page 5 in the revised manuscript.

**5.** *Figure 1: This can be more legible and well distributed. The mesh and bathymetry info can be in one figure. The figure is not indicating anything like region, the Sea etc. Including that can be a good idea. Figure 1. (a) can be an inset image and other information are important could be enlarged. A legible north arrow with map scales, coordinates lat & long N, E degree etc are also recommended (this comment is applicable for all maps).*

**AR:** As indicated in the response to specific comment 1, the Figure 1 (page 4 in the revised manuscript) is modified for better clarity including all suggestions. The authors wish to keep the bathymetry and mesh as separate figures and regions in the map are indicated and also the legends are corrected as suggested. The correction is applied for all maps as suggested.

**6.** *Give the legend names (eg. Figure 7, wave height (m) than mentioning meter in legend ranges), x- and y-axis variables in each plots, etc*

**AR:** As suggested all figures are checked and necessary changes are made in the legend and labels wherever appropriate. The revised figures are inserted in the revised manuscript.

**7.** *Line 199: Mention which ECMWF wind: ERA5, ERA interim etc.*

**AR:** As mentioned everywhere in the manuscript, we use ECMWF analysis winds (see line 140, page 6 in the revised manuscript) not reanalysis. We added in the caption of Fig. 4 (line 270, page 12 in the revised manuscript) ECMWF analyses specification.

**8.** *Appreciate explaining why the zones or control points are assigned? Or on what basis?*

**AR:** The zones are divided based on the various characteristics such as trophic conditions, prevailing local wind-wave characteristics. The control points are the standard locations along the Emilia-Romagna coastal belt where regional agencies such as Arpae carry out measurements. Hence the same has been used in the present study.

We added the following comment at new line 102-105, page 5:

The ER coastal area is subdivided into three major zones (Fig. 1c) which correspond to different coastal trophic conditions (Fiori et al., 2016). The station locations are the land town locations perpendicular to which the environmental agency monitoring transects are done monthly and weekly to monitor the marine ecosystem good environmental status. Thus, knowing the prevailing winds and waves at these locations could be of importance for the management of this important coastal area.

9. *Line 156: Why there is no ST6+ JONSWAP (EXP4)?*

**AR:** The authors understand this comment and would like to clarify that a combination of ST6+ JONSWAP is not considered because already JONSWAP bottom friction showed less performance in EXP1.

We added the following comment at line 192-193, page 8 in the revised manuscript:

The combination of ST6 with JONSWAP is not considered because this bottom friction is not suitable for sandy beaches as already the EXP1 will show.

10. *Line 157: Apt to mention why the representative months are February and September?*

**AR:** The months of February and September are considered mainly representing the seasons namely winter and autumn.

These changes appear in line 192, page 8 in the revised manuscript.

11. *Section 3.3: Why the validation was done for each year separately? What is the significance of that?*

**AR:** A clarification is offered. The comparison of significant wave height is shown for each year in Fig. 2 (Page 9) so as to have an immediate check on the reproduction of wave heights by the model at the buoy location over the years 2010-19. This is sometimes called *"consistency check"* in meteorological literature which we now mention.

We added the following sentence at line 209-210, page 8 in the revised manuscript:

This is a consistency check of model against observations as required for "goodness" indicators in numerical weather predictions (Murphy, 1993).

12. *Figure 3 (f-j): It might be suitable to use 45-degree line than best fit line to best understand the underestimation and bias in validation. Check for wave characteristic notations in graph axis too.*

**AR:** As suggested the line of no deviation (1:1 slope) is added in the Figure 3 (page 11, in the revised manuscript). The notation in the graph axis is checked and corrected. The figure legend is corrected (lines 248-249, page 11 in revised manuscript).

13. *Sub-section 4.1.1 is the only subsection under 4.1, which can be merged with section 4.1 itself.*

    **AR:** The authors wish to retain the section heading as such.

14. *Section 4.1.1: The wind speed values seems to be lower than a desired value. And SD is more than the annual mean wind speed which is not right. Check those values for wind speed everywhere. The mean value can come around 5 m/s to 20 m/s to produce the wave characteristics indicated in the MS. Make changes in text and figures accordingly. I recommend comparing these values with in-situ measurement if you have any for authors clarity.*

    **AR:** The authors have responded to this comment in the **general comments (a).** Please see above.

15. *Figure 4 &6, mark degree N, E on Latitude and Longitude coordinate axis*

    **AR:** As suggested the Figure 4 & 6 (Pages 12 & 14 in the revised manuscript) axis labels are corrected.

16. *Revise Figure 5 according to new wind speed data and try to correlate Figure 5 and Figure 7 for any influence of predominant wind direction on wave characteristics.*

    **AR:** The authors would like to clarify again that there is no error in the wind data as responded in the **general comments (a)**.

17. *Line 285: Sticking to one notation of position can help the reader understand better. Either 'control points' or 'stations. Indicating 'point' can sometimes make confusions.*

    **AR:** The control points (1-5,7,8) refer to the locations considered along the Emilia-Romagna coastal belt where the wind-wave characteristics was analyzed. But the location 6 alone is referred as station because it the coastal buoy location where the validation was executed. It is to be made clear that other 7 control points are not stations but are points chosen by authors for the wind-wave analysis in the present study.

18. *What is the significance of analysis of $25^{th}$ day of month and monthly mean seasonal spectra for each year? What are the concluding remarks of these analysis could be mentioned?*

    **AR:** The $25^{th}$ day of the month is **chosen to show an example and it just our choice**.

    The instantaneous spectra are presented to show the co-existence of sea-swell characteristics in the study domain. Unlike the averaged spectra the instantaneous spectra provide a clear picture of the sea-swell dominance characteristics during the period 2010-19.

    The monthly mean seasonal spectra are considered for each year to have an idea about the evolution of wave spectra for the years respective of each season. This gives us an idea about the spectral shapes as well as the spectral features (single/double-peaked) prevalent in the coastal belt. Note that there is no other study that has reported on the wave spectra characteristics in the Emilia Romagna coastal belt. The limitation being the non-availability of wave spectra measurements for comparison.

For better clarity the authors analyzed the percentage of occurrence of single and double-peaked spectra for the various seasons for the 10 years. The conclusion from this analysis is included in the lines 390-396 (Page 18 in the revised manuscript) and in the summary & conclusions section (lines 551-553, page 25). Table 5, indicating the number of occurrences of single-peaked, double-peaked, and multi-peaked spectra at Cesenatico location in different seasons (2010-19) is also added (Page 19, line 400 of the revised manuscript).

**19.** *Line 315: 'As seen from the Fig.???' to be filled*

**AR:** The Figure number is added as suggested.

**20.** *Line 425: 'the comparison of Tm and Tp...... 10 years' needs clarity. Please reframe by adding adequate information.*

**AR:** The sentence is corrected for better clarity (line 540, page 25 in the revised manuscript).

**21.** *A discussion on limitations/uncertainties of this study could be added. Such as limitations of Weibull distribution.*

**AR:** The author comment is well appreciated, and limitations are added in the conclusions. This has been stated in the revised manuscript (Pages 25 & 26, lines 563-568 in the revised manuscript) and as shown below.

The limitation in the present study is the non-availability of wave spectra measurements at the coastal locations for validation. Future study will aim to consider data assimilation (de Rosnay et al., 2022) and to have higher resolutions winds as forcings for the wave model. In the context of heavy tailed data sets the Weibull distribution may not represent a best description of the peak and tail. These limits can be overcome by adding more parameters such as the 4-parameters exponentiated Weibull distribution (Mahmoudi et al., 2018) such that the extra shape parameter can provide more versatility to the distribution in the shape of the tails.

**Technical /minor corrections**

    **i.**     *Line 13: 'direction' to 'wave direction'*

        **AR:** The correction is made as indicated.

    **ii.**     *Significant wave height could be H subscript s. This applies with every wave characteristic. Using the global notations can be beneficial for a wider audience in understanding this research more.*

        **AR:** The authors have used uniform abbreviations to represent the significant wave parameters mentioned in the text such as significant wave height (Hs), Mean wave period (Tm), peak wave period (Tp), and mean wave direction ($\theta$m) which are mostly commonly used ones. Hence the authors wish to retain it in the present form.

**iii.** *Expand acronym at first appearance will be appreciated. Few eg: are line 26 IPCC, WW3 is not expanded anywhere in MS, Line 101: 'JONSWAP parameterization (Joint North Sea Wave Project)', this can be 'Joint North Sea Wave Project (JONSWAP) parameterization', Explain ST4, ST6, CFL etc. Could use the acronyms after defining. PDF is expanded twice and is using the same.*

**AR:** The acronym expansions at its first appearance is checked and corrected as pointed out (IPCC, WW3, CFL, PDF). In the case of JONSWAP and SHOWEX the authors prefer to indicate the expansion in brackets as they are well known formulations. ST4 and ST6 are wind-input dissipation parametrizations, and they are mentioned the same way as indicated in the WW3 manual. Repetitive mention of PDF is corrected as suggested.

**iv.** *Line 121: check '~4.5 km hourly'*

**AR:** The typo is corrected.

**v.** *After defining Significant wave height ($H_S$) use the same in everywhere.*

**AR:** The same has been responded in technical/minor corrections, comment (ii).

**vi.** *I recommend the authors to make the decimal places consistent in entire MS. Eg. Line 126 location coordinates has 4 decimal places. Most research work would go for 2 decimal numbers.*

AR: The location coordinates are corrected to two decimal places as suggested. While in the tables indicating the statistics the authors wish to retain up to third decimal places and hence the same is retained.

**vii.** *Line 180: 'measurement's' to 'measurements'*

**AR:** The correction is made as indicated.

**viii.** *Line 188: 'Fig. 3' to 'Figure 3'*

**AR:** The correction is made as indicated.

**ix.** *Line 190: 'relatively a good' to 'relatively good'*

**AR:** The correction is made as suggested.

**x.** *Line 250: 'and northern' to 'and the northern'*

**AR:** The correction is made as suggested.

**xi.** *Line 327: 'upon blowing of the wind' to 'wind characteristics'*

**AR:** The sentence is corrected as indicated.

**xii.** *Line 348: 'costal' to 'coastal'*

**AR:** The typo error is corrected.

***xiii.***     *Line 423: 'A Hs' to 'An'*

       **AR:** The typo error is corrected.

***xiv.***     *Line 435: 'waves' to 'wave'*

       **AR:** The correction is made as suggested.
* * *
**Responses to Reviewer 2:**

**Dear Reviewer,**

We thank you for the thorough evaluation of our manuscript. The constructive comments and suggestions have improved the manuscript highly. Below, we address each of your comments in turn. The comment is repeated in italics and our authors' response (AR) directly follows highlighted in red color. The AR contains our reply and a brief description of what has been modified in the manuscript. In the revised manuscript, the corrections suggested have been addressed.

**Best Regards,**
**Umesh (on behalf of all co-authors)**

**General comments**

*The manuscript represent an important work and a substantial contribution to the understanding of natural hazards on ER area by the use of new methods like the hazard index. The used data and tools are up to international state of the art.*

*The scientific approaches and the applied methods are valid and discussed in an appropriate and balanced way. There are many references considering previous works on the area. The scientific methods and assumptions are clearly presented and can be reproduced thanks to the codes in annexe. The discussion could be broadened by a point on wind quality which is detailed below in this review.*

*The scientific data and results are precisely and clearly illustrated and presented with appropriate figures and tables. Rare remarks below are aimed to document still wider the results.*

**AR:** Thank you for your in-depth and detailed evaluation of our manuscript. The authors have noted the reviewer's suggestions and comments. The corrections suggested by reviewer #2 are incorporated into the revised version of our manuscript.

**Specific comments**

(a) *The forcing by a 6-hourly wind at 0,125° on a area with such an important space and time variability than the Adriatic Sea seems to be the first limit of the wave climatology. Several results show it : first of all, the degraded scores on summer. Indeed, you show the worse correlation and bias of SWH in this season. The wave model understimate the wave height of 11 cm. It is known that global atmospheric model at such a scale don't represent explicitly the convection and are not appropriate to simulate surface wind due to heating flux that occurs at this season. It is consistent with the lack of energy in the wave model compared to the observation.*
*This limit also appear through the small standard deviation in summer. Is the standard deviation of the observation so weak than in the model during this season ? A part of this decrease is due to the summer low wave height, but it could also come partially from a poor representation of convective wind.*
*I would recommend to better document this limit by some elements, for instance :*

*- a comparison with an observed wind climatology, even by a station on land. It could also be the climatology from a mesoscale model with hourly time step. Does it present the same standard deviation of wind speed in summer than the experiment, the same wind rose ? Are the relative quality of wind different between the seasons ?*

*- more explorative : what is the error in SWH or period depending on the wind direction observation (or from a mesoscale model) ? Indeed the global ECMWF model may have specific limitation for some local wind.*

*Moreover, a mention of this limit should appear in the discussion. Even in winter a 6-hourly wind has consequences on the model wave quality. Mediterranean sea is known for its very dynamic storms.*

**AR:** The reviewer comment is well appreciated and a clarification is offered.

We present a validation of wind speed and direction from available data sets for the locations of Porto Corsini, Porto Garibaldi and Cesenatico Port, locations are given in Table 1.

**Table 1:** Details of coastal stations used for validation of wind forcing

| Station | Lat (°N) | Lon (°E) |
|---|---|---|
| Porto Corsini | 44.49 | 12.28 |
| Porto Garibaldi | 44.67 | 12.24 |
| Cesenatico Port | 44.20 | 12.40 |

[Figure]

**Figure I:** Comparison of ECMWF winds with measurements at the Porto Corsini station for the year 2013: (a) wind speed (m/s), and (b) wind direction (deg.).

Figure I (a, b) shows the comparison of wind speed and direction with the observations at Porto Corsini for the year 2013. The ECMWF winds show an overall consistency with observations. The comparison statistics for all three stations is shown in Table 2. We see that overall, the correlation at all stations and seasons is higher than 0.7 with the exception of summer and autumn, as expected because of intense air-sea interaction in the coastal boundary layer which are not captured by the ECMWF model. The Bias is low while the root mean square error between

observations and ECMWF is about 2 m/s in all seasons. This is visualized also with the wind speed scatter plots in Figure-II.

**Table 2:** Quality assessment of ECMWF winds with observed wind speeds for selected stations (refer Table 1).

| Statistics | Wind speed (m/s) | | | | |
|---|---|---|---|---|---|
| | (a) Porto Corsini [Year: 2013] | | | | |
| | Full year | Winter | Spring | Summer | Autumn |
| R | 0.7 | 0.7 | 0.7 | 0.5 | 0.7 |
| Bias | -0.2 | 0.2 | -0.1 | -0.3 | -0.4 |
| RMSE | 1.8 | 1.8 | 1.9 | 1.6 | 2 |
| (b) Porto Garibaldi [Year: 2018] | | | | | |
| R | 0.7 | 0.8 | 0.7 | 0.5 | 0.8 |
| Bias | -0.2 | 0.2 | -0.3 | -0.5 | 0 |
| RMSE | 1.8 | 1.7 | 1.6 | 1.9 | 1.9 |
| (c) Cesenatico Port [Year: 2015] | | | | | |
| R | 0.7 | 0.8 | 0.8 | 0.5 | 0.6 |
| Bias | -0.2 | 0 | -0.3 | -0.6 | 0.2 |
| RMSE | 1.9 | 1.7 | 2 | 1.9 | 2 |
| *R: Correlation, RMSE: Root Mean Square Error* | | | | | |

[Figure]

**Figure II:** Scatter plot of wind speed (m/s) at the three stations **(a)** Porto Corsini, **(b)** Porto Garibaldi, and **(c)** Cesenatico port. The red dotted lines represent the best data fit and the black dotted lines indicates the 1:1 slope. *[R: Correlation, B: Bias, and RMSE: Root Mean Square Error].*

Thus, we argue that in this region ECMWF wind quality is reasonable. Naturally in the future higher resolution limited area modelling winds could be used to reinforce the characterization of the wind climatology in this area but at this time, long time series are not available.

We added the following explanation at line 140-144 (Page 6 in the revised manuscript), along with Table 1 (line 146, page 6 in the revised manuscript) indicating the wind speed validation statistics in the revised manuscript:

The model winds were validated at three stations, namely Porto Corsini (44.49°N, 12.28°E), Porto Garibaldi (44.67°N, 12.24°E) and Cesenatico Port (44.20°N, 12.40°E) along the ER coastal belt. The wind speed comparison statistics as indicated in Table 1 showed correlations of the order 0.7, with bias of -0.2 m/s indicative of underestimation of wind speed, and RMSE of 1.8 m/s. Larger

biases of the order of -0.6 m/s and correlations as low as 0.5 exist during summer and some autumn seasons.

Furthermore, in the conclusions section we added a comment about the limitation of the present study due to the low-resolution winds. The comment is added in lines 569-573, page 26 in the revised manuscript:

Another limitation is that a 10-year period would generally not be enough to bring out the complete climatological wave response to winds. ECMWF winds were too low resolution before 2010 and no downscaled limited area meteorological forcing is available. Hence this 10-year period could be the first reference database for the prevailing wind-wave characteristics in the coastal belt for researchers and coastal engineers/designers. Future works definitely should deal with longer simulation periods and also higher resolution winds.

**(b)** *l.151 Sensitivity experiments for wave model parametrizations: no test of different parameters value of each physics has been conducted, at least there is no mention of it in the paper. The validation of the method by 3 model configurations seems all the same sufficient. But the conclusion could be less affirmative regarding the better capacity of ST6 to represent sea state on ER area. We could think that another parametrisation of ST4 than the one in Ardhuin et al. 2010 could have produced better results than EXP3.*

*So I advice to moderate the sentence l.421-422 (Summary and conclusions) in a way like : « The sensitivity tests has shown the good accuracy of ST6+SHOWEX physics for wave hindcasts in the study area. »*

**AR:** As suggested by the reviewer the sentence is corrected as shown below. These changes appear in lines 536-537, page 25 in the revised manuscript.

The sensitivity tests has shown the good accuracy of ST6+SHOWEX physics for wave hindcasts in the study area.

**(c)** *l.160 and 191: It would be very interesting to add the mean value of the observation for the sensitivity periods and the whole 10 years. Thus the reader would be aware of the relative error at this point.*

**AR:** Thank you for pointing out this. As suggested the mean values of the observation for the sensitivity periods and the validation period (2010-19) is indicated in the respective sections for better clarity. These changes appear in line 199 (page 8) and lines 228-229 (page 10) in the revised manuscript.

**(d)** *l.267: Is it possible to add a comparison against the observed wave rose of the station 6?*

**AR:** The authors very well agree with the reviewer that it would be good to have observed wave rose of the station 6 and hence based on available data for the Cesenatico station the wave roses for selected years are compared with the model estimates as shown in Figure III.

[Figure]

**Figure III:** Comparison of directional histograms of wave heights: buoy (a, c, e), and simulated data (b, d, f) at the station 6 (Cesenatico).

These changes including Figure 8 (page 16 in the revised manuscript) showing the comparison against the observed wave rose of the station 6 are presented in lines 354-361, page 16 in the revised manuscript and as shown below.

Based on available buoy data for the Cesenatico station the observed wave roses are compared with the model estimates for selected years as shown in Fig. 8. Overall, the modelled wave roses (Figs. 8b, d, f) shows a reasonable correspondence with the observed data (Figs. 8a, c, e), even with some difference in magnitudes. An underestimation of model wave heights in the lower ranges is noted. Comparing the directional distributional of waves, the directions are comparable being in the same sectors but there exist higher differences in their magnitudes. Similar wave climate by the Nausicaa buoy located offshore of Cesenatico is reported in studies by Armaroli et al. (2012), and Romagnoli et al. (2021), which shows that this is the representative wave climate of the Emilia-Romagna coast. This qualitative comparison shows that at the Cesenatico station overall characteristics of waves are fairly reproduced by the model.

*(e) l.304: The bimodality in winter and summer doesn't seem so obvious on the graphics. In winter, I consider visually 3 cases of double peaked spectra. In summer, it is more complicated to distinguish and I don't see a lot more than 1 case. Thus majority of cases appear to be single peaked.*

*I would recommend either to write clearly the number or proportion of cases that are bimodal by season in order to attest it solidly, or to pursue the exploration of data by adding more cases. Indeed all the days of February (August) of the 10 years could be examined. The proportion of bimodal cases could then be adressed on a significant number of occurrences.*

*If the results show effectively a proportion of around 30% of bimodal spectra on these 2 seasons, I would nuance the conclusion. For instance, it would be appropriate to write « during winter and summer the spectra have often/sometimes bimodal characteristics », than « during winter and summer the spectra have bimodal characteristics » (l304-305).*

**AR:** The reviewer comment is well appreciated and for better clarity the authors analyzed the model instantaneous wave spectra data for the complete period of 2010-19. The seasonal percentage of occurrence of single, double-peaked and multi-peaked spectra is presented in Table 3 as shown below.

**Table 3.** Number of occurrences of single-peaked, double-peaked, and multi-peaked wave spectra at Cesenatico location for different seasons (2010-19).

| Seasons (2010-19) | Single-peak (%) | Double-peak (%) | Multi-peak (%) |
|---|---|---|---|
| Winter | 31 | 45 | 24 |
| Spring | 32 | 45 | 23 |
| Summer | 27 | 53 | 20 |
| Autumn | 33 | 49 | 18 |

The conclusion from this analysis (as shown below) are included in the lines 390-396 (Page 18 in the revised manuscript) and in the summary & conclusions section (lines 551-553, page 25). Table 5, indicating the number of occurrences of single-peaked, double-peaked, and multi-peaked spectra at Cesenatico location for different seasons (2010-19) is also added (Page 19, line 400 of the revised manuscript).

The spectra vary considerably over the years and in general, the wave spectra at the Cesenatico coastal location showed signatures of single and double-peaked spectra for the period 2010-19 (Table 5). The wave spectra were prominently double-peaked during all seasons (45-53 %), along with single-peaked spectra but with a lesser percentage of occurrences (27-33 %). Double peakedness was highly prominent in summer season (53%), while winter, spring and autumn showed dominance of single-peaked spectra (31-33 %). As evident from Table 5, the percentages of the number of peaks (single/ double) in the Cesenatico location clearly depicts the co-existence of sea-swell characteristics in the study domain.

**Technical corrections**

**(i) l.24:** *« is crucial » instead of « is a crucial »*

**AR:** The correction is made as indicated.

**(ii) l.30:** *« Other state-of-the-art models include » instead of « Other state-of-the-art models includes »*

**AR:** We have applied the reviewer suggestion.

***(iii) l.89:*** *the formulation isn't very clear. Proposition : « Armaroli et al. (2012) reported that waves originating from east correspond to a proportion of 91%HS < 1,25 m, owing to the controlled fetch. »*

**AR:** We revised the sentence as suggested by the reviewer (line 110, page 5 of revised manuscript).

***(iv) l.93:*** *« action-density » without space.*

**AR:** The typo error is corrected as pointed out.

***(v) l.109:*** *« The model spectrum is sampled in 24 directions and 30 frequencies (0,0500-0,7932 Hz), with an increment factor of 1.1.*

**AR:** The sentence is reframed as suggested by the reviewer (line 131, page 6 of revised manuscript).

***(vi) l.113:*** *ST4 isn't mentioned there, wich is a bit confusing. Indeed ST4 and ST6 are both introduced earlier, and are actually used in the validation. It would be clearer to add a mention here of JONSWAP and ST4, precising that they were used only for the validation.*

**AR:** Thank you for the comment. We realized that repetition was not needed so we deleted the phrase with ST6 in this position.

***(vii) l.176:*** *Please indicate in the legend of the table 3 the number of used buoys to be sure that it takes into account only the station 6 or the whole control points.*

**AR:** The legend of Table 3 (i.e Table 4 in the revised manuscript) is corrected as shown below for better clarity (line 226, page 10 in the revised manuscript).

"Statistics of the comparison of buoy measurements (Cesenatico, Station 6) with model results for 2010-2019".

***(viii) l.388:*** *The chosen value of Xc could appear in the legend of Table 4. Its value appears only in page 20.*

**AR:** The Xc value is added in the legend of Table 4 (i.e. Table 6 in the revised manuscript) as indicated below (line 502-503, Page 23 in the revised manuscript):

"Table 6. The best-fit Weibull scale and shape parameters for Hs (columns 3-4) at the eight control points. Column 5 shows the estimated $\chi^2$ values. Mean, variance, skewness, and kurtosis of the Hs (columns 6-9) computed from the model data (left sub-columns) and from the Weibull fit parameters (right sub-columns), and the wave height hazard index (with threshold value Xc, Hs=1.08m) calculated with (eq. 6) for the eight control points (indicated in column 10) along the Emilia-Romagna coastal strip".

***(ix) l.418:*** *I would suppress the upper-case C of Conclusions, except if asked by the editor.*

**AR:** The typo is corrected as suggested.
* * *